# Effect of a Novel Trivalent Vaccine Formulation against Acute Lung Injury Caused by *Pseudomonas aeruginosa*

**DOI:** 10.3390/vaccines11061088

**Published:** 2023-06-11

**Authors:** Keita Inoue, Mao Kinoshita, Kentaro Muranishi, Junya Ohara, Kazuki Sudo, Ken Kawaguchi, Masaru Shimizu, Yoshifumi Naito, Kiyoshi Moriyama, Teiji Sawa

**Affiliations:** 1Department of Anesthesiology, Graduate School of Medical Science, Kyoto Prefectural University of Medicine, Kyoto 602-8566, Japan; keitaino@koto.kpu-m.ac.jp (K.I.); mao6515@koto.kpu-m.ac.jp (M.K.); j-ohara@koto.kpu-m.ac.jp (J.O.); a080025@koto.kpu-m.ac.jp (K.S.); ken1113@koto.kpu-m.ac.jp (K.K.); masaru@koto.kpu-m.ac.jp (M.S.); ynaitoh@koto.kpu-m.ac.jp (Y.N.); 2Department of Emergency and Critical Care Medicine, Faculty of Medicine, Fukuoka University, Fukuoka 814-0180, Japan; muranishi@adm.fukuoka-u.ac.jp; 3Department of Anesthesiology, School of Medicine, Kyorin University, Mitaka 181-8611, Japan; mokiyo@ks.kyorin-u.ac.jp

**Keywords:** bacterial components, infection immunity, toxin, pathogenesis, trivalent vaccine

## Abstract

An effective vaccine against *Pseudomonas aeruginosa* would benefit people susceptible to severe infection. Vaccination targeting V antigen (PcrV) of the *P. aeruginosa* type III secretion system is a potential prophylactic strategy for reducing *P. aeruginosa*-induced acute lung injury and acute mortality. We created a recombinant protein (designated POmT) comprising three antigens: full-length PcrV (PcrV_#1-#294_), the outer membrane domain (#190-342) of OprF (OprF_#190-#342_), and a non-catalytic mutant of the carboxyl domain (#406-613) of exotoxin A (mToxA_#406-#613(E553Δ)_). In the combination of PcrV and OprF, mToxA, the efficacy of POmT was compared with that of single-antigen vaccines, two-antigen mixed vaccines, and a three-antigen mixed vaccine in a murine model of *P. aeruginosa* pneumonia. As a result, the 24 h-survival rates were 79%, 78%, 21%, 7%, and 36% in the POmT, PcrV, OprF, mTox, and alum-alone groups, respectively. Significant improvement in acute lung injury and reduction in acute mortality within 24 h after infection was observed in the POmT and PcrV groups than in the other groups. Overall, the POmT vaccine exhibited efficacy comparable to that of the PcrV vaccine. The future goal is to prove the efficacy of the POmT vaccine against various *P. aeruginosa* strains.

## 1. Introduction

*Pseudomonas aeruginosa*, a gram-negative aerobic bacillus, rarely causes serious infections in healthy individuals but causes opportunistic infections in immunocompromised patients [1,2]. In immunocompromised patients, *P. aeruginosa* is the causative agent of skin infections following severe burns and ventilator-associated pneumonia in intensive care settings. Since the 1970s, research has been conducted on immunotherapy using vaccines as an effective prophylactic means to replace antibacterial drug therapy [3,4]. Multidrug-resistant *P. aeruginosa* has emerged in recent years [5,6,7], but active and passive immunotherapy that does not rely on antibacterial agents for *P. aeruginosa* infection has not yet reached clinical application.

To date, we have demonstrated the active immunization effect of V antigen (PcrV) targeting the type III secretion system involved in *P. aeruginosa* virulence, as well as passive immunization using a specific antibody (anti-PcrV antibody) that blocks the PcrV antigen [8]. *P. aeruginosa* uses its type III secretion system to inject four type III secretory toxins (ExoS, ExoT, ExoU, ExoY) into target eukaryotic cells, thereby exerting pathogenic toxicity [9]. Among these toxins, ExoU, a phospholipase toxin, causes cell damage and significant pathogenic toxicity in conditions associated with epithelial cell damage, such as acute lung injury and corneal ulcers [9,10]. PcrV is the structure at the tip of the secretion needle of the type III secretion system, and the PcrV antibody strongly binds to the tip of this secretion needle, thereby inhibiting the translocation of type III secretory toxins to the target eukaryotic cells [11,12,13].

Although it is clear from past studies that acute lung injury, septic pathology, and lethality in *P. aeruginosa* pneumonia are highly dependent on the pathogenicity of the type III secretion system, it has been reported that pathogenic factors other than the type III secretion system are involved in the pathogenesis of *P. aeruginosa* infection. They have been explored as target antigens [1,2]. In active immunotherapy against *P. aeruginosa* infection, vaccine therapy using the outer membrane protein OprF and extracellular protein exotoxin A as antigens is also being studied globally. OprF is an outer membrane protein expressed in all *P. aeruginosa* serotypes. It has been utilized as a recombinant OprF/I fusion protein in both active and passive immunization [14,15,16]. Exotoxin A is a type II secretion system cytotoxin reported to be produced by 70%–90% of *P. aeruginosa* strains [17,18,19]. It functions as an ADP-ribosyl transfer toxin that causes cell death by inhibiting protein synthesis [20]. Since *P. aeruginosa* has various virulence factors and causes various infectious diseases, a multivalent vaccine targeting multiple antigens affecting major virulence factors is ideal for an anti-*P. aeruginosa* vaccine. For example, vaccines against pathogenic bacteria, such as pneumococcal vaccines, have been developed to be polyvalent vaccines effective against various pneumococcal subspecies [21]. Similarly, because *P. aeruginosa* has a variety of pathogenic mechanisms and causes many different types of infections, there has always been the question of whether a monovalent vaccine, such as the PcrV vaccine alone, would be able to cover the various aspects of *P. aeruginosa* infection. From this viewpoint, a trivalent *P. aeruginosa* vaccine was prepared and evaluated for protective efficacy against acute *P. aeruginosa* lung injury in this study. In addition to the type III secretion system PcrV, we combined the domain of the outer membrane protein OprF and the enzymatic domain of non-catalytically mutated exotoxin A to prepare an antigen protein (POmT) vaccine antigen. Our past animal studies have shown that active immunization with recombinant PcrV can significantly reduce acute pulmonary epithelial injury that occurs and exacerbates within 4 h following a lethal dose of *P. aeruginosa* pulmonary infection. Consequently, active immunization with PcrV can suppress subsequent pulmonary edema and inflammation, improve bacterial clearance in the lung, and reduce acute mortality within 24 h. Therefore, using our murine model in the current study, we evaluated the preventive effect of the newly created three-antigen POmT against *P. aeruginosa*-induced acute lung injury by comparing the prophylactic effects against recombinant PcrV and two additional antigens.

## 2. Materials and Methods

### 2.1. Construction of Recombinant Tagless Proteins

The coding sequences of full-length PcrV, the outer membrane domain (#190-342) of OprF (OprF_#190-#342_), and the carboxyl domain (#406-613) of exotoxin A (ToxA_#406-#613_) were polymerase chain reaction (PCR)-amplified from the *P. aeruginosa* PA103 chromosome. Specific PCR primer sets are listed in Appendix A. The enzymatic active glutamic acid at #553 of toxA_#406-#613_ was deleted by a specific PCR oligonucleotide (toxA_D553) to generate a non-catalytic mutated exotoxin A fragment (mToxA_#406-#613(E553Δ)_) [22,23,24]. The DNA sequences of PcrV, OprF_#190-#342_, and mToxA_#406-#613(E553Δ)_ were connected to glycine–serine short linker oligonucleotide sequences (5′-GGA TCC GCC ACC GCC-3′ for GGGGS between PcrV and OprF_#190-#342_, and 5′-ACC GGA ACC CCC TGA ACC-3′ for GSGGSG between OprF_#190-#342_ and mToxA_#406-#613(E553Δ)_) to create a synthetic trivalent protein antigen named POmT (which stands for a conjugate of PcrV and parts of OprF and mutated exotoxin A) composed of full-length PcrV-GGGGS-OprF_#190-#342_-GSGGSG-mToxA_#406-#613(E553Δ)_. The PCR amplicons were ligated to the cloning site of an expression vector (32932, TAGZyme pQE-2, Qiagen, Hilden, Germany) to create recombinant hexahistidine-tagged proteins in *Escherichia coli*. After the purification, dialysis, and endotoxin removal processes, the amino-terminal hexahistidine tags of the recombinant proteins were removed using exopeptidases (34362, TAGZyme DAPase Enzyme, Qiagen, Hilden, Germany). The above details were described in our past report [25]. Tagless POmT, composed of 666 amino acids with two glycine–serine linkers, was a 72.22-kDa protein. The blueprint of each protein antigen is presented in Figure 1.

### 2.2. Immunizations

Certified pathogen-free male ICR mice (4 weeks old; body weight, 20–25 g) (Shimizu Laboratory Supplies, Co., Ltd., Kyoto, Japan) were housed in cages (with filter tops) under pathogen-free conditions. All animal experiments were performed under the approval of the Animal Research Committee of the Kyoto Prefectural University of Medicine (Approval numbers: M2019-563, M2020-314, M2021-335, M2022-326). One set of experiments consisted of 10–15 mice from three vaccination groups, and several sets were repeated on different days to check the reproducibility of the effects of vaccination.

The recombinant proteins (10 µg/dose), used at a previously reported effective antigen dose [8], were formulated in combination with aluminum hydroxide gel (alum, 20 mg/mL, 100 µL/dose, Alhydrogel adjuvant 2%, vac-alu-250, InvivoGen, Toulouse, France). Aluminum hydroxide gel solution was added into the solution containing one vaccine protein or a combination of multiple proteins at a final volume ratio of 1:1 and was mixed well by pipetting the solution up and down for at least 5 min. The solution was diluted to 200 µL/dose with saline. On day 0, eight groups of mice were vaccinated subcutaneously on their backs with the vaccine adjuvant conjugate. In addition, one group of mice was injected with alum alone (Table 1). Twenty-eight days later, booster immunizations or injections were administered using the same formulations.

### 2.3. Anti-PcrV, Anti-OprF, Anti-ToxA, and Anti-POmT Titer Measurements

On 22 days after the final immunization (On day 50 from the first immunization), in the vaccinated mice, the peripheral blood was collected to determine specific IgG titers to the vaccine proteins by enzyme-linked immunosorbent assay (ELISA), as reported previously [25].

### 2.4. Infectious Challenge by P. aeruginosa

The *P. aeruginosa* PA103, originally from an Australian patient [26], is a cytotoxic strain with positive type III secretion of ExoT and ExoU. PA103*ΔpcrV, a* mutant lacking *the pcrV* gene, was previously produced by the Dara W. Frank laboratory (Department of Microbiology and Immunology, Medical College of Wisconsin) [8]. Bacterial suspensions were prepared, as reported previously [25,27]. On day 56 (28 days after the final immunization, 12 weeks old), the solution containing either PA103 (1.0 × 10^6^ cfu in 60 µL of saline) or PA103*ΔpcrV* (1.0 × 10^8^ cfu in 60 µL of saline) was instilled into the lung of each vaccinated mouse through an endotracheal needle, as described previously under light inhalational anesthesia using sevoflurane [27]. PA103*ΔpcrV* was administered at a 100-fold higher dose than PA103. Mice in the saline (no infection) group were given saline intratracheally only. The survival and body temperatures of all mice were monitored for 24 h, after which the survivors were euthanized. As we reported previously, the mice which received a lethal dose of *P. aeruginosa* PA103 (1.0 × 10^6^ cfu, LD_50%_~24 h) became hypothermic within 4 h [8,10]. The hypothermic pathology and mortality within 24 h in this mouse model of pneumonia is not a result of bacterial multiplication in the infected organ but a consequence of cell necrosis of lung epithelial cells due to the translocation or the type III secretory toxin ExoU of *P. aeruginosa* into the lung epithelial cells within 4 h after infection [28]. We have also reported that the necrosis of lung epithelial cells due to ExoU toxin translocation occurs within 1 h after infection in an in-vitro culture cell model [10]. In addition, it has been reported that this pathology disappears after infection with an isogenic mutant strain lacking the enzymatically active ExoU of *P. aeruginosa* [10]. Therefore, our prophylactic model focuses on the first 4 to 24 h after infection. Accordingly, the experiment was completed in 24 h to focus on the pathogenesis of acute lung injury induced by *P. aeruginosa*, in addition to considerations on animal welfare and preventing contamination within the facility according to the infection experiment regulations of our facility. It is worth noting that the reasons for the different numbers of animals in each group are as follows. In a single experiment, a maximum of 15 mice were used, including the POmT group as a positive control (effective) group and the alum group as a negative control (ineffective) group, with other groups to be investigated and interposed in between to determine the effect. Data were obtained for more than 10 animals/group, with experiments repeated at least three times to confirm reproducibility. Therefore, in total, the PA103 infection series comprised 42 animals in the POmT group and 28 animals in the alum group, which were both larger than the other groups (10–18 animals). The lungs of each mouse were collected and homogenized using a homogenizer (Polytron PT10/35, Kinematica, Luzern, Switzerland) for further evaluation.

### 2.5. Lung Edema Index, Myeloperoxidase (MPO) Activities, and Bacteriological Assay

The wet-to-dry weight ratio of the lungs was measured as an index of lung edema at the 24-h time point, as described previously [28]. MPO activity in the lung homogenates was measured, as reported previously [25]. The sequentially diluted lung homogenate was inoculated on a sheep blood agar plate and incubated at 37 °C overnight to calculate the number of remaining bacteria in a gram of lung tissues.

### 2.6. Histopathological Assay

Lung tissue was collected at 24 h after infection from surviving mice, two mice per group, and subjected to histological evaluation. The lungs were perfused with 10% formalin neutral buffer solution for fixation and embedded in paraffin. The mounted sections were stained with hematoxylin–eosin.

### 2.7. Statistical Analysis

For multiple comparisons of survival curves, the post-hoc analysis for the log-rank test was used with *p*-values adjustment by the Benjamini & Hochberg method using RStudio (2022.7.1. Build 554, R version 4.2.1, RStudio PBC). One-way analysis of variance and Bonferroni’s test (Prism 9, GraphPad Software, La Jolla, CA, USA) were used to compare IgG titers, body temperatures, the lung edema index, and the MPO assay data. The non-parametric tests (Kruskal–Wallis test) and the multiple comparison tests (Dunn’s test) were used to compare the number of remaining lung bacteria with a statistical significance (*p*-value < 0.05).

## 3. Results

According to the vaccination protocol (Figure 2), in which mice were vaccinated using a three-antigen conjugated vaccine (POmT), one-antigen vaccine (PcrV, OprF, or mTox), two-antigen- mixed vaccine (PcrV+OprF, PcrV+mTox, or OprF+mTox), three-antigen mixed vaccine (PcrV+OprF+mTox), or saline with alum alone, we determined the preventive effect against pulmonary infection by *P. aeruginosa* PA103, which strongly expresses type III secretory toxicity, or the mutant strain PA10*3ΔpcrV*, which has lost type III secretory toxicity because of deletion of the *pcrV* gene.

### 3.1. Specific Antibody Titers Induced by Vaccination

In the first series of experiments in which vaccinated mice were eventually infected with *P. aeruginosa* PA103 (Figure 3a), serum anti-POmT IgG titers in mice vaccinated against POmT, PcrV, OprF, or mTox were statistically significantly increased compared with the control titers († *p* < 0.01). Anti-PcrV IgG titers from mice vaccinated against POmT or PcrV, anti-OprF IgG titers from mice vaccinated against POmT or OprF, and anti-mTox IgG titers from mice vaccinated against POmT or mTox were significantly increased († *p* < 0.01). In the second series of experiments, in which vaccinated mice were later infected with the isogenic mutant PA103Δ*pcrV* strain (Figure 3b), vaccinated mice exhibited a similar antibody titer profile as those in the first series of experiments.

### 3.2. Survival Rates and Body Temperatures

All groups of mice, except the saline-treated control mice, were infected intratracheally with PA103 (1.0 × 10^6^ cfu) or PA103*ΔpcrV* (1.0 × 10^8^ cfu) on day 56 after the first immunization, and their survival and body temperature were monitored for 24 h (Figure 4).

#### 3.2.1. Comparison between the POmT and Monovalent Vaccines

First, the POmT group was compared with the one-antigen vaccine (PcrV, OprF, or mTox) groups. The 24-h survival rates were 79%, 78%, 21%, 7%, and 36% in the POmT, PcrV, OprF, mTox, and alum-alone groups, respectively (Figure 4a). Statistically, the POmT and PcrV group had significantly higher survival rates than the alum-alone group († *p* < 0.001 and † *p* = 0.019, respectively). The survival rate in the mTox group was significantly lower than those in the alum alone († *p* = 0.012) and PcrV (‡ *p* = 0.0002) groups. The body temperature in surviving mice 12 h after PA103 administration was 34.3 ± 2.2 °C in the POmT group, 33.7 ± 2.3 °C in the PcrV group, 33.6 ± 2.7 °C in the OprF group, 29.6 ± 2.4 °C in the mTox group, and 31.1 ± 3.1 °C in the alum-alone group. Body temperatures at 12 h significantly differed between the POmT and alum-alone groups († *p* < 0.001). There was no statistically significant difference in survival or body temperature between POmT and PcrV groups.

#### 3.2.2. Comparison of POmT and Bivalent Combination Vaccines

Second, the two-antigen mixed vaccine groups (PcrV+OprF, PcrV+mTox, or OprF+mTox) were evaluated (Figure 4b). The 24-h survival rate was 70% in the PcrV+OprF group, 67% in the PcrV+mTox group, 9% in the OprF+mTox group, 79% in the POmT group, and 36% in the alum-alone group. The survival rates were significantly higher in the PcrV+OprF and PcrV+mTox groups than in the OprF+mTox group (‡ *p* = 0.006 and ‡ *p* = 0.006, respectively). The survival rate in the OprF+mTox group was significantly lower than in the POmT group (‡ *p* < 0.0001). Thus, retention of the PcrV antigen component in the vaccine was critical for protective efficacy. At 12 h after PA103 administration, the body temperature of the surviving mice was 34.8 ± 2.0 °C in the PcrV+OprF group (§ *p* = 0.013 vs. OprF+mTox), 33.7 ± 2.4 °C in the PcrV+mTox group, and 28.2 ± 0.2 °C in the OprF+mTox group. Thus, the presence of the PcrV antigen from the vaccine was critical for preventing infectious hypothermia following PA103 infection.

#### 3.2.3. Comparison of POmT and the Trivalent Combination Vaccine

Third, the three-antigen mixed vaccine (PcrV+OprF+mTox) group was compared with the POmT group (Figure 4c). There was no significant difference in the survival rate 24 h after PA103 administration between these groups (79% vs. 70%). At 8 h after bacterial administration, the body temperature in the surviving mice was 32.2 ± 2.3 °C in the POmT group and 32.9 ± 1.8 °C in the PcrV+OprF+mTox group. At 12 h after infection, the body temperature was 34.3 ± 2.2 °C in the POmT group and 34.3 ± 1.5 °C in the PcrV+OprF+mTox group. At 12 h, the body temperature was significantly higher in the POmT and PcrV+OprF+mTox groups than in the alum-alone group († *p* < 0.001 and † *p* = 0.016, respectively). However, body temperature did not differ between the POmT and PcrV+OprF+mTox groups over 24 h.

#### 3.2.4. Comparison between the POmT and PcrV Monovalent Vaccine Groups following Infection with *P. aeruginosa* PA103ΔpcrV

The PA103Δ*pcrV* strain lacks type III secretory toxicity because of an artificial genetic defect in PcrV. Therefore, pulmonary injury and death following pulmonary infection by PA103*ΔpcrV* are attributable to pathogenicity other than type III secretory toxicity. However, the lethal dose of the PA103*ΔpcrV* strain in a mouse lung injury model is approximately 100-fold higher than that of the wild-type strain PA103. Therefore, we compared the effects of the POmT and PcrV vaccines on the survival of mice infected with the PA103*ΔpcrV* strain (Figure 4d). The survival rate of PcrV-vaccinated mice infected with PA103Δ*pcrV* was 47%, which was not significantly lower than that of uninfected mice (*p* = 0.23), while the survival rates in the alum and POmT groups were 50% and 68%, respectively. Although the survival rate with POmT was higher than that with PcrV, this difference was not statistically significant (*p* = 0.23). The survival rates of PA103 (1 × 10^6^ cfu)-infected mice treated with POmT, PcrV, and alum were 79%, 78%, and 36%, respectively, while the survival rates of PA103Δ*pcrV* (1 × 10^8^ cfu)-infected mice treated with POmT, PcrV, and alum were 68%, 47%, and 50%, respectively. Although the bacterial dose of PA103Δ*pcrV* was 100 times higher than that of PA103, there was no statistically significant difference in mortality between PA103-infected and PA103Δ*pcrV*-infected mice and the corresponding vaccine groups.

### 3.3. Lung Edema, MPO Assay, Bacteriology, and Lung Histology

The severity of lung edema was measured in mice infected with the PA103 and PA103*ΔpcrV* strains (Figure 5a). Following PA103 administration, all mice exhibited severe lung edema compared with uninfected control mice (3.3 ± 0.2, * *p* < 0.002). The lung edema index was 5.4 ±1.4 in the POmT group and was the highest in the OprF+mTox group (6.6 ± 0.8), although there was no statistically significant difference among the groups infected with PA103. There were also no significant differences among the POmT, PcrV, and alum-alone groups following *P. aeruginosa* PA103*ΔpcrV* administration.

The MPO activities at 24 h in *P. aeruginosa*-infected lungs were quantified to evaluate the severity of inflammation (Figure 5b). The alum-alone, POmT, PcrV, OprF, and mTox groups showed significant increases in MPO activity compared with the uninfected group (* *p* < 0.0001–0.008). Among the PA103-infected groups, MPO activity was significantly lower in the POmT and PcrV groups than in the alum-alone group († *p* < 0.0001 and † *p* = 0.002, respectively).

Total bacterial counts in the lungs were calculated in both surviving and dead mice 24 h after *P. aeruginosa* infection (Figure 6). Following PA103 infection, lung bacterial counts were significantly lower in the PcrV+OprF+mTox group than in the alum-alone group (Figure 6a) († *p* = 0.006). No significant differences were observed between the other groups and the alum-alone group. Following PA103*ΔpcrV* infection, the number of bacteria was lower in the POmT group than in the PcrV and alum groups, albeit without statistical significance (Figure 6b). More mice in the group immunized with POmT were found to have a decreased number of bacteria in their lungs relative to the initial bacterial dose (cfu/gram of lung tissue) (§ *p* < 0.05) (red dashed lines, Figure 6a).

Histological changes were evaluated in the mouse lungs 24 h post-infection. Destruction of the alveolar structure was observed with enhanced neutrophil infiltration and alveolar hemorrhage in the lungs of mice in the alum-alone, OprF, and mTox groups (Figure 7). On the other hand, the lungs of POmT-vaccinated mice and PcrV-vaccinated mice displayed significantly fewer inflammatory changes than those of the other groups. It is assumed that lung histology showed less inflammation than the average for the group because lung tissue was collected from surviving mice over 24 h in each group. These histological findings roughly correlated with the MPO assay results.

## 4. Discussion

Over nearly half a century, several *P. aeruginosa* virulence factors that are potential vaccine antigens have been identified and investigated as candidate vaccine formulations. However, all vaccine trials using these virulence factors have ended without satisfactory results that could lead to clinical use [4]. OprF/I, an outer membrane protein, is highly immunogenic and expressed by all *P. aeruginosa* serotypes. A vaccine containing this outer membrane protein (IC43) exhibited a strong safety profile in a preclinical study, eliciting antibodies that promote complement-dependent opsonization of *P. aeruginosa*, leading to a high degree of efficacy against *P. aeruginosa* infection [29]. Based on its protective effect, it was advanced to Phase III trials [30,31]. Although good immunogenicity and safety were observed, the mortality and infection rates showed no difference from those in the placebo group. Based on its potent cytotoxicity, exotoxin A was identified as a cytotoxic factor in the pathogenicity of *P. aeruginosa* and was proposed as a candidate vaccine antigen. However, few studies have tested exotoxin A alone as a vaccine antigen. Both passive and active immunization were previously used for antigens in which exotoxin A is conjugated to LPS, and neither approach was successful [32,33,34]. However, vaccines against exotoxin A have not been applied clinically to date.

The type III secretion system of gram-negative bacteria accomplishes the direct delivery of protein toxins from bacterium to host by nanosyringe “injectisomes,” which form a conduit across the two bacterial membranes, extracellular space, and the plasma membrane of a target eukaryotic cell [9]. PcrV, which has a cap-like structure at the tip of the needle-like structure of the *P. aeruginosa* injectisome, is involved in the toxin translocation mechanism across the plasma membrane of a target eukaryotic cell [11,12,13]. In our previous research, we found that a specific antibody against PcrV suppressed the toxicity of the type III secretion system [11]. A clinical trial of this antibody has been conducted, but it has not yet been put to practical use [35,36,37]. Because *P. aeruginosa* carries a variety of virulence factors, studies have been conducted on vaccines focusing on each virulence factor, but none have been applied clinically. Recently, the effectiveness of vaccines (active immunization) with bivalent or trivalent antigens has been investigated [38,39,40,41] to broaden the spectrum of prophylactic efficacy against *P. aeruginosa* subspecies by targeting multiple virulence factors rather than a single virulence factor. To develop a multivalent *P. aeruginosa* vaccine, inhibition of the type III secretion system, the major virulence factor of *P. aeruginosa*, is of utmost importance. Based on these reports, we focused on the pathogenic diversity of *P. aeruginosa*, including PcrV. We decided to create a multivalent vaccine, which included the following components: (1) The type III secretion system PcrV of *P. aeruginosa* exerts pathogenicity by injecting toxin proteins into target cells using the secretion apparatus; (2) OprF is an outer membrane protein expressed in all *P. aeruginosa* serotypes and is effective in both active and passive immunization; (3) The type III toxin secretion system causes cell death by inhibiting protein synthesis in target cells, and 70%–90% of clinical strains are reported to be exotoxin A-producing strains [18,19]. We compared the active immunization efficacy of a novel vaccine POmT conjugated with three monovalent antigens using a monovalent vaccine containing the conventional monovalent vaccines PcrV, OprF, and mTox and a mouse model of *P. aeruginosa* pneumonia.

In this study, significant differences were observed between the POmT vaccine group and the alum-alone group (control group) regarding the 24-h survival rate after PA103 strain infection and the change in body temperature 12 h after infection. In contrast, there was no significant difference in any variable between the POmT vaccine and any vaccine containing PcrV. These findings indicate that the binding of OprF and exotoxin A to PcrV does not eliminate the vaccine effect of PcrV, that is, the inhibitory effect of the type III secretion system, and the POmT vaccine antigen is active against acute lung injury caused by *P. aeruginosa*. In addition, in comparison to serum IgG, it was revealed that antibody titers against OprF and exotoxin A increased steadily following immunization with the novel tri-antigen POmT vaccine to the same level as achieved with immunization by individual protein antigens. However, immunization with OprF alone, mTox alone, and OprF+mTox resulted in a lower survival rate than that in the alum group. It remained unknown whether immunization with OprF or mTox brought about an adverse immune effect on survival or whether the lower survival rate was due to the instability of the experiment. However, the OprF, mTox alone, and bivalent OprF+mTox vaccines did not demonstrate the positive immune effects of PcrV and POmT.

The aforementioned results using *P. aeruginosa* strain PA103 confirmed that the type III secretion system is significantly involved in acute lung injury caused by *P. aeruginosa*. To evaluate the protective effect of the POmT vaccine against pathogens other than the type III secretion system, we conducted an infection experiment using the *pcrV* deletion mutant PA103*ΔpcrV*, which lacks a functional type III secretion system and compared it with the POmT vaccine [8]. No significant difference was observed between the POmT and PcrV vaccine groups. These findings indicate that the pathogenic mechanism of acute lung injury induced by *P. aeruginosa* is highly dependent on the type III secretion system and is largely unrelated to other pathogenic factors. Previous studies on vaccine efficacy against *P. aeruginosa* LPS and OprF were planned before the discovery of the *P. aeruginosa* type III secretion system and ExoU toxin, an effector molecule representing its toxicity, in the mid-1900s. As our results suggest, the ability to counteract *P. aeruginosa* type III secretion system-induced toxicity is essential for preventing and treating acute pulmonary *P. aeruginosa* infections. However, it remains possible that vaccines can be effective against *P. aeruginosa* infectious pathologies dependent on other virulence factors. At least in the present study, the POmT vaccine showed significantly higher survival rates in a lethal *P. aeruginosa* pneumonia model, suggesting that the POmT vaccine has the potential to be effective in preventing a more comprehensive range of *P. aeruginosa* infections. Finally, in this experiment, all prepared component vaccines were administered subcutaneously to evaluate the protective effect of the POmT vaccine with those administered using more conventional immunization methods. *P. aeruginosa* is a pathogen that infects the respiratory tract, so airway mucosal immunity is essential. As we recently reported the efficacy of intranasal administration of the PcrV-CpG deoxyoligonucleotide vaccine [25], our future studies will focus on confirming the effectiveness of intranasal administration of the POmT vaccine.

Since our discovery in 1999 of the immune effect of the PcrV antigen against *P. aeruginosa* pneumonia [8], we have been working on developing PcrV-targeted therapeutic antibodies and PcrV vaccines. However, since *P. aeruginosa* retains diverse pathogenic mechanisms and causes various infectious diseases [5], many pharmaceutical companies have discouraged the development of immunotherapies targeting only a single PcrV antigen, mainly because of its potential narrow spectrum against various *P. aeruginosa* pathogenicities. No vaccine against *P. aeruginosa* has been clinically applied to date [4]. Since there is no evidence that a multi-antigen vaccine is superior to a PcrV single-antigen vaccine in *P. aeruginosa* infection, we investigated this by comparing vaccines targeting only PcrV with vaccines that include other antigens. Unfortunately, as described above, POmT, a trivalent vaccine, failed to show a definite advantage over the mono-antigen PcrV vaccine. However, the findings of this experiment revealed several important points. First, we confirmed that *P. aeruginosa* type III secretory toxicity and its inhibition are critical factors in the etiology and prevention of *P. aeruginosa*-induced acute lung injury, with other virulence factors playing a minor role. Second, the three antigens, artificially linked via an amino-acid linker in the molecular structure, retained the antigenicity and immune effects of PcrV. This finding was significant because, in producing vaccines containing three separate components, manufacturing regulations require that safety and efficacy tests are conducted for each element, including antigens and adjuvants [42], complicating the development of multi-antigen vaccines and raising costs considerably. Therefore, combining antigens at the gene level to generate a single molecule is considered the most realistic option when manufacturing a multi-protein antigen vaccine. As mentioned above, there is a concern that vaccines that do not contain PcrV (OprF and/or mTox), at least in our *P. aeruginosa* pneumonia model, may exacerbate rather than improve lung injury. In the case of immunotherapies against virus infections, the possibility that immunization against specific antigens that do not suppress toxicity exacerbates cellular damage, especially of phagocytic cells, has been reported as the antibody-dependent enhancement (ADE) phenomenon [43]. This mechanism of ADE involves the interaction of pathogen–antibody immune complexes to phagocytic cells through the adherence of the antibody Fc region with cellular Fc receptors. In this phenomenon, non-neutralizing antibodies generated by vaccination and bound to pathogen proteins can promote the subsequent contact of pathogens to the target cells and intensify the inflammatory process during infection. Therefore, antibodies derived from vaccines that do not suppress cytotoxicity may promote the interaction of pathogens with phagocytic cells and eventually induce more severe cytotoxicity. A mechanism similar to the ADE phenomenon may help to explain the poor performance of the present PcrV-free OprF and/or mTox vaccine.

## 5. Conclusions

This study created POmT as a single recombinant protein by combining PcrV, OprF, and mTox with glycine-serine linkers. In terms of preventing acute lung injury and subsequent death due to *P. aeruginosa* infection, immunization with POmT demonstrated comparable prophylactic levels with PcrV immunization according to all tested parameters, such as specific antibody titers, survival, lung edema, MPO, bacteriology in the lungs, and lung histology. In addition, the protective effects of the POmT vaccine were almost the same as those observed with the mixture of PcrV, OprF, and mTox. In our mouse model series, we could not demonstrate the benefit of including OprF and/or mTox as vaccine components against *P. aeruginosa* pneumonia. We confirmed that immunity to PcrV is essential for preventing *P. aeruginosa*-induced acute lung injury and that immunity to OprF and exotoxin A has little effect. In addition, when using genetic recombination technology, even with the artificially synthesized trivalent vaccine, antibody titers increased. Moreover, the impact on PcrV, without affecting the antigenicity of the components, including PcrV, was confirmed. Whether the *P. aeruginosa* trivalent vaccine POmT is effective against other *P. aeruginosa* strains and different *P. aeruginosa* infections is an issue that will be addressed in future studies.

## Figures and Tables

**Figure 1 vaccines-11-01088-f001:**
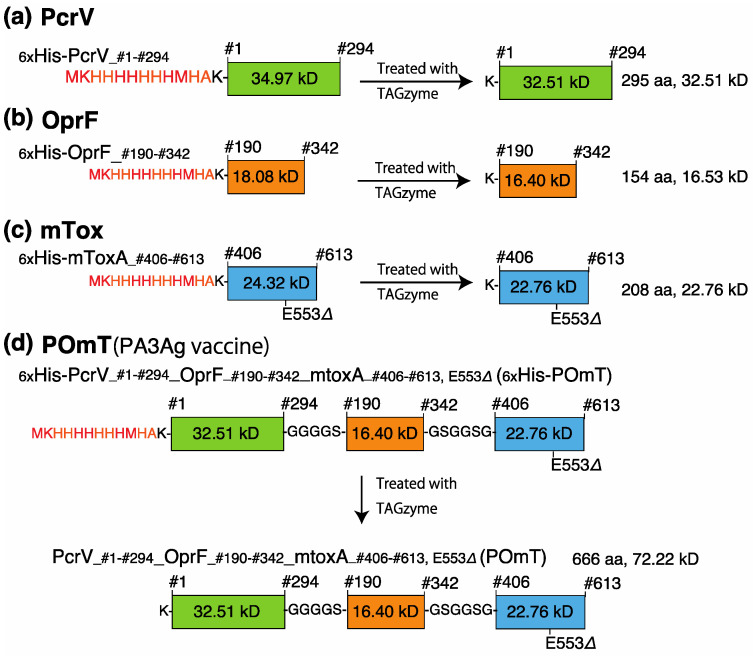
The vaccine designs of recombinant protein antigens. Four recombinant proteins, namely (**a**) PcrV, (**b**) OprF, (**c**) mTox, and (**d**) POmT, were designed as vaccine antigens for this project. The coding sequences of full-length PcrV, the outer membrane domain (#190-342) of OprF. (OprF_#190-#342_), and the carboxyl domain (#406-613) of exotoxin A (ToxA_#406-#613_) were polymerase chain reaction (PCR)-amplified from the *P. aeruginosa* PA103 chromosome. Specific PCR primer sets are listed in Appendix A. The enzymatic active glutamic acid at #553 of toxA_#406-#613_ was deleted by a specific PCR oligonucleotide (toxA_D553) to generate a non-catalytic mutated exotoxin A fragment (mToxA_#406-#613(E553Δ)_). The DNA sequences of PcrV, OprF_#190-#342_, and mToxA_#406-#613(E553Δ)_ were connected to glycine–serine short linker oligonucleotide sequences to create a synthetic trivalent protein antigen designated POmT (which stands for a conjugate of PcrV and parts of OprF and mutated exotoxin A). The PCR amplicons were ligated to the protein expression vectors. After the expression and purification of recombinant proteins, the amino-terminal hexahistidine tags of the proteins were removed using exopeptidase.

**Figure 2 vaccines-11-01088-f002:**
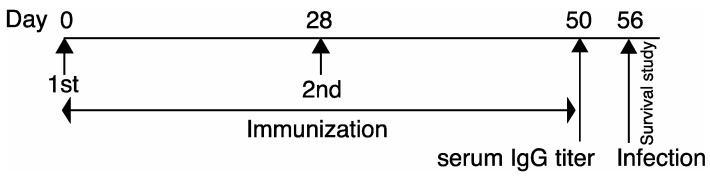
Experimental protocol. Mice were subcutaneously immunized twice on their back on days 0 and 28. On day 50, a small volume of peripheral blood was collected from the tail vein of each mouse, and the specific titers of vaccine antigens were measured in each serum sample. Then, on day 56, each mouse underwent tracheal instillation of *P. aeruginosa* PA103 (1.0 × 10^6^ cfu/mouse) or its *pcrV*-deleted isogenic mutant PA103Δ*pcrV* (1.0 × 10^8^ cfu/mouse), and their body temperature and survival were monitored for 24 h.

**Figure 3 vaccines-11-01088-f003:**
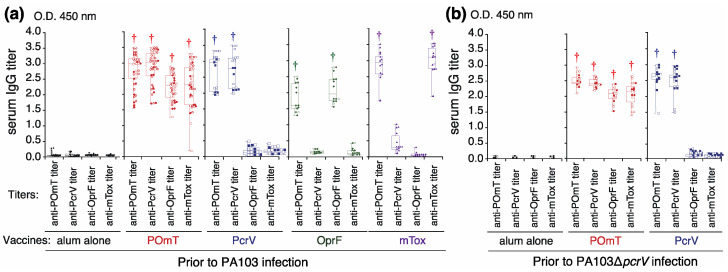
Specific IgG titers in the sera of the vaccinated mice. The titers were measured 50 days after the initial immunization. Anti-POmT, anti-PcrV, anti-OprF, and anti-mTox titers were measured in mice vaccinated with alum alone, POmT, PcrV, OprF, or mTox before challenge with either PA103 or PA103Δ*pcrV*. (**a**) Specific IgG titers (anti-POmT, anti-PcrV, anti-OprF, and anti-mTox) in the sera of mice with vaccinated alum alone, POmT, PcrV, OprF, or mTox before intratracheal infection with PA103. (**b**) Specific IgG titers in the sera of mice with vaccinated alum alone, POmT, or PcrV before intratracheal infection with PA103Δ*pcrV*. Data are presented as the median (a center solid bar in the box) and the first (boxes) and second quartiles (bars), with each value indicated by symbols (open symbol: survived, close symbol: died). † *p* < 0.05 compared with the alum-alone group. PcrV, full-length PcrV; OprF, the outer membrane domain (#190-342) of OprF; mTOX, a non-catalytic mutated domain (#406-613, E553Δ) of exotoxin A; POmT, a conjugate of PcrV, and parts of OprF and mutated exotoxin A.

**Figure 4 vaccines-11-01088-f004:**
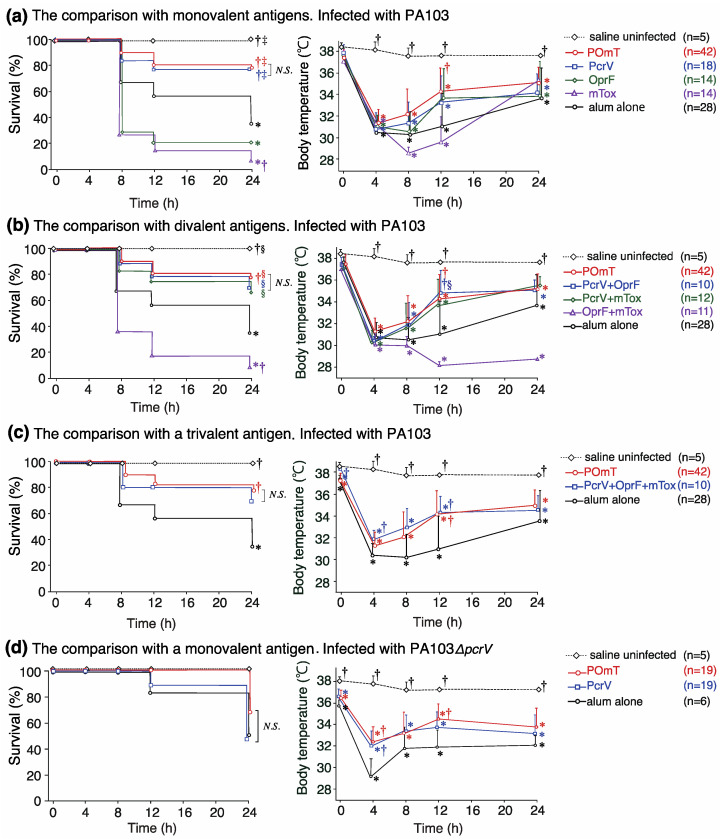
Survival rates (%) and body temperature (°C) over time in vaccinated mice for 24 h after intratracheal infection with either PA103 or PA103ΔpcrV. (**a**) Mice were vaccinated with monovalent antigens (POmT, PcrV, OprF, mTox, or alum alone) and intratracheally infected with PA103. (**b**) Mice were vaccinated with divalent antigens (PcrV+OprF, PcrV+mTox, or OprF+mTox) and intratracheally infected with PA103. (**c**) Mice were vaccinated with trivalent antigens (PcrV+OprF +mTox) and intratracheally infected with PA103. (**d**) Mice were vaccinated with a single-protein antigen (POmT, PcrV, or alum alone) and intratracheally infected with PA103ΔpcrV. Data are shown as the mean ± SD. * *p* < 0.05 against the saline uninfected group. † *p* < 0.05 agasint the alum-alone groups. ‡ *p* < 0.05 agasint the mTox group. § *p* < 0.05 compared with the OprF+mTox group. N.S., not significant. PcrV, full-length PcrV; OprF, the outer membrane domain (#190-342) of OprF; mTOX, a non-catalytic mutated domain (#406-613, E553Δ) of exotoxin A; POmT, a conjugate of PcrV, and parts of OprF and mutated exotoxin A.

**Figure 5 vaccines-11-01088-f005:**
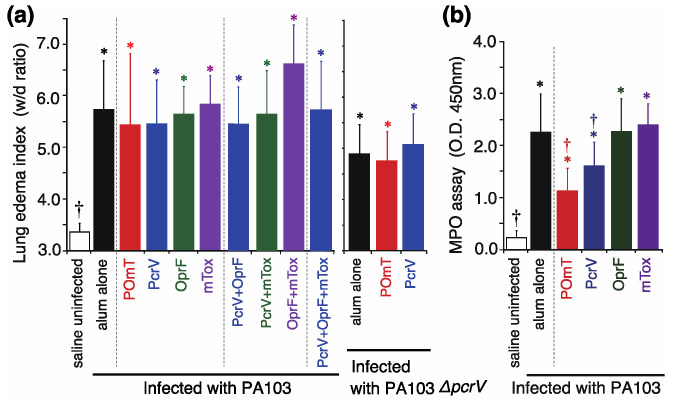
Lung edema index and myeloperoxidase (MPO) activities in lung homogenates from vaccinated mice infected with *P. aeruginosa*. (**a**) Lung edema index at 24 h after intratracheal infection with either *P. aeruginosa* PA103 or PA103*ΔpcrV*. (**b**) MPO activities in the lung 24 h after infection with PA103. Data are shown as the mean ±SD. * *p* < 0.05 against the saline-treated uninfected control; † *p* < 0.05 against the alum-alone-vaccinated group. PcrV, full-length PcrV; OprF, the outer membrane domain (#190-342) of OprF; mTOX, a non-catalytic mutated domain (#406-613, E553Δ) of exotoxin A; POmT, a conjugate of PcrV, and parts of OprF and mutated exotoxin A; MPO, myeloperoxidase; w/d, wet-to-dry weight ratio.

**Figure 6 vaccines-11-01088-f006:**
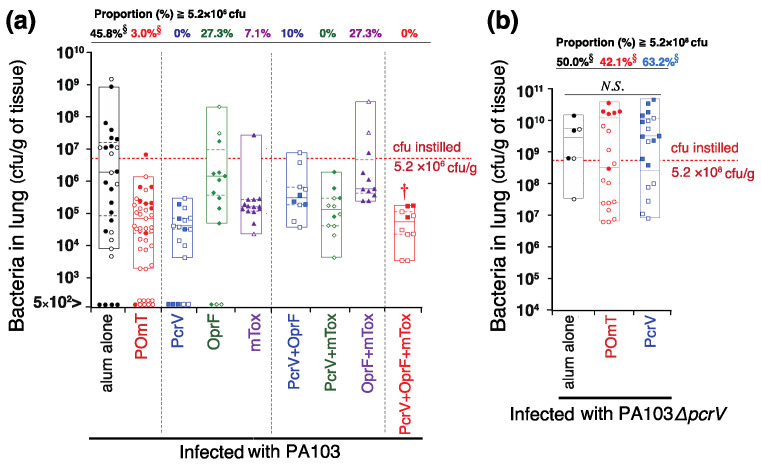
Bacterial colony-forming units (cfu) in lung homogenates from the vaccinated mice infected with *P. aeruginosa*. Bacterial cfu in mouse lungs 24 h after intratracheal infection with *P. aeruginosa.* (**a**) PA103-infected groups. (**b**) PA103*ΔpcrV*-infected groups. Data (cfu/gram of lung tissue) are presented as the median (a center solid bar in the box) and the first (dotted lines in the box) and 5th & 95th percentiles (boxes), with each value indicated by symbols (open symbol: survived, closed symbol: died). † *p* < 0.05 compared with the alum-alone-vaccinated group. Proportion (%): the percentage of mice in which the number of bacteria remaining in the lung was ≥ the initial bacterial dose. § *p* < 0.05 for the rate (%) of occurrence of a greater number of remaining bacteria in the lung than the initial bacterial dose (5.2 × 10^6^ cfu/gram and 5.2 × 10^8^ cfu/gram of normal lung tissue for PA103 and PA103*ΔpcrV*-infection, respectively). Red dashed lines demonstrate the initial bacterial dose. PcrV, full-length PcrV; OprF, the outer membrane domain (#190-342) of OprF; mTOX, a non-catalytic mutated domain (#406-613, E553Δ) of exotoxin A; POmT, a conjugate of PcrV, and parts of OprF and mutated exotoxin A.

**Figure 7 vaccines-11-01088-f007:**
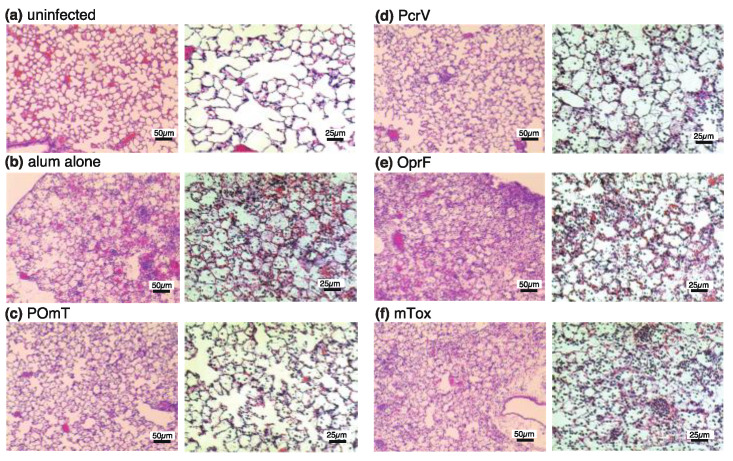
Lung histology of the surviving mice (two mice/group) at 24 h post-infection with *P. aeruginosa* PA103. After the fixation with 10% formaldehyde, hematoxylin–eosin staining and paraffin embedding were performed. (**a**) Saline-treated uninfected group, (**b**) alum-alone group, (**c**) POmT group, (**d**) PcrV group, (**e**) OprF group, and (**f**) mTox group. Magnification 200×, scale bar = 50 µm. Magnification 400×, scale bar = 25 µm. PcrV, full-length PcrV; OprF, the outer membrane domain (#190-342) of OprF; mTOX, a non-catalytic mutated domain (#406-613, E553Δ) of exotoxin A; POmT, a conjugate of PcrV, and parts of OprF and mutated exotoxin A.

**Table 1 vaccines-11-01088-t001:** Vaccination groups.

Group	n	Antigen	Adjuvant
saline uninfected	5	none	-
Infected with PA103			
alum alone	28	none	alum
POmT	42	POmT	alum
PcrV	18	PcrV	alum
OprF	14	OprF_#190-#342	alum
mTox	14	ToxA__#406-#613(E553Δ)_	alum
PcrV+OprF	10	PcrV+OprF___#190-#342	alum
PcrV+mTox	12	PcrV+ToxA__#406-#613(E553Δ)_	alum
OprF+mTox	11	OprF___#190-#342+ToxA__#406-#613(E553Δ)_	alum
PcrV+OprF+mTox	10	PcrV+OprF__#190-#342_+ToxA__#406-#613(E553Δ)_	alum
Infected with PA103Δ*pcrV*			
alum alone	6	none	alum
POmT	19	POmT	alum
PcrV	19	PcrV	alum

PcrV, full-length PcrV; OprF, the outer membrane domain (#190-342) of OprF; mTOX, a non-catalytic mutated domain (#406-613, E553Δ) of exotoxin A; POmT, a conjugate of PcrV, and parts of OprF and mutated exotoxin A.

## Data Availability

The datasets generated and/or analyzed in this study are available from the corresponding author upon reasonable request.

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
