# Peer review of "Effect of a Novel Trivalent Vaccine Formulation against Acute Lung Injury Caused by Pseudomonas aeruginosa"

_vaccines, 2023, doi:10.3390/vaccines11061088_

Round 1

Reviewer 1 Report

The manuscript by Inoue et al., describes experiments designed to examine the protective effect of a recombinant P. aeruginosa vaccine.  The vaccine (POmT) consists of a full length PcrV protein, the outer membrane domain of OprF, and the non-catalytic mutant of the carboxy terminus domain of exotoxin A. The efficacy of the vaccine was compared with that of: a) single protein vaccine (PcrV, OprF, and mToxA); b) two antigen mixed vaccine (PcrV + OprF, PcrV + mToxA, and OprF + mToxA); and three antigen mixed vaccine (PcrV+ OprF + mToxA). The efficacy of the vaccine was compared/assessed using the murine model of acute lung infection and the P. aeruginosa strain PA103.  The 24-hour survival rate was 79%, 78%, 21%, 7%, and 36% for mice vaccinated with POmT, PcrV, OprF, mToxA, and alum alone (control) respectively. In addition, mice vaccinated with either POmT or PCRV showed a significantly faster recovery from hypothermia.  Based on these results, the authors suggested that the efficacy of POmT is comparable to that of PcrV one.  In addition, the authors suggested that efficacy of POmT against various P. aeruginosa strains may be assessed.

Critique:

It is clear that the authors invested considerable time and efforts conducting the described experiments and preparing the manuscript.  The manuscript is well-written and the description of the results is clear and the interpretations are justified.  However, there are several potential problems:

1)      The main problem relates to the experimental design and the stated goal of the study.  It appears that the authors planned to determine if an additive improvement in the vaccine is achieved by including PcrV, mToxA, or both.  Ii is already known that PcrV vaccine produces a considerable protection. Clinical trials including PcrV monoclonal antibodies were conducted.  This potential additive effect of OprF, mToxA, or both should be stated very clearly in this manuscript.  In addition, and as stated above, the authors compared several combinations.  However, their abstract described the comparison of POmT with either PcrV, OprF, or mToxA only. Furthermore, the conclusions clearly indicate that no significant improvement was achieved by including either OprF or mToxA to the PcrV vaccine.  This is important as the authors stated a goal of proving the efficacy of POMT against other P. aeruginosa strains.  How about testing the original PcrV vaccine?  It has been tested against other P. aeruginosa strains.  In conclusion, the goal should be a comparative analysis by including either OprF or mToxA to the observed protection.

2)      Figure 4-a: The survival rate among mice vaccinated with PMOT and PCRV was 79% and 78% respectively.  Is 78% significant but 78% not? (statistically speaking).

3)      Figure 4-a: It is intriguing that the survival rate among mice vaccinated with alum was higher than those vaccinated with either OprF or mToxA (36% vs. 21% and 7%).

4)      Figure 4, a and b: It is intriguing that the survival rate of the combined vaccine (OprF + mToxA) is 9% whereas that of the single OprF is 21%. Does this suggest that mToxA lessen the efficacy of the OprF?

5)      Figure 4d: Survival analysis using PA103DpcrV: The authors compared the survival rate of mice vaccinated with either the trivalent vaccine or PCRV vaccine.  Since earlier experiments with the wt. PA103 (figure 4, a-c) clearly showed that PcrV is the most efficient, one would expect the authors to compare the groups vaccinated with OprF, mToxA, or a combination of those. PCRV vaccination against PA103Dpcrv will not lead to the required important comparison.  To delineate the contribution of OprF, mToxA, or both, the mutant lacks PcrV but still produces OprF and toxin A.

6)      Possibly the authors need to provide a comparison of the survival rate among mice infected with either PA103 and PA103DpcrV (with or without vaccination).  This helps the readers appreciate the contribution of PcrV or the T3SS through PCRV.

7)      Figure 5-a, the lung edema index: The authors clearly showed that the index in mice vaccinated with POMT is significantly lower than those vaccinated with OprF-mToxA (53% vs. 66%).  However, the index in mice vaccinated with OprF or mToxA a is also low (55% and 58%).  The comparison should include all types of vaccinations.

8)      Figure 6, lung bacterial load: a) the authors stated that the lung bacterial counts were significantly lower in the POMT, PCVR, and PcrV + OprF + MToxA than mice treated with alum alone.  No significant deference was detected between mice treated with other groups and alum.  However, if one expects the observed significant reduction is due to the presence of PcrV, what about the combined treatment with PcrV + OprF or PcrV + mToxA? Both contain PCRV; b) the authors reported the lung bacterial load as CFU/lung.  However, variations likely to occur.  This should be standardized.  The reporting should be: CFU/gm of lung tissue.

9)      The backbone of this study is the trivalent vaccine (POmT).  Therefore, a possible section for the comparison of POMT with the PCRV + OPRF + MToxA vaccine is essential.  Is there a difference in any of the tested categories?

no concern

Author Response

Responses to the reviewer #1 comments

Reviewer #1

The manuscript by Inoue et al., describes experiments designed to examine the protective effect of a recombinant P. aeruginosa vaccine.  The vaccine (POmT) consists of a full length PcrV protein, the outer membrane domain of OprF, and the non-catalytic mutant of the carboxy terminus domain of exotoxin A. The efficacy of the vaccine was compared with that of: a) single protein vaccine (PcrV, OprF, and mToxA); b) two antigen mixed vaccine (PcrV + OprF, PcrV + mToxA, and OprF + mToxA); and three antigen mixed vaccine (PcrV+ OprF + mToxA). The efficacy of the vaccine was compared/assessed using the murine model of acute lung infection and the P. aeruginosa strain PA103.  The 24-hour survival rate was 79%, 78%, 21%, 7%, and 36% for mice vaccinated with POmT, PcrV, OprF, mToxA, and alum alone (control) respectively. In addition, mice vaccinated with either POmT or PCRV showed a significantly faster recovery from hypothermia.  Based on these results, the authors suggested that the efficacy of POmT is comparable to that of PcrV one.  In addition, the authors suggested that efficacy of POmT against various P. aeruginosa strains may be assessed.

Critique:

It is clear that the authors invested considerable time and efforts conducting the described experiments and preparing the manuscript.  The manuscript is well-written and the description of the results is clear and the interpretations are justified.  However, there are several potential problems:

The main problem relates to the experimental design and the stated goal of the study.  It appears that the authors planned to determine if an additive improvement in the vaccine is achieved by including PcrV, mToxA, or both.  Ii is already known that PcrV vaccine produces a considerable protection. Clinical trials including PcrV monoclonal antibodies were conducted.  This potential additive effect of OprF, mToxA, or both should be stated very clearly in this manuscript.  In addition, and as stated above, the authors compared several combinations.  However, their abstract described the comparison of POmT with either PcrV, OprF, or mToxA only. Furthermore, the conclusions clearly indicate that no significant improvement was achieved by including either OprF or mToxA to the PcrV vaccine.  This is important as the authors stated a goal of proving the efficacy of POMT against other P. aeruginosa strains.  How about testing the original PcrV vaccine?  It has been tested against other P. aeruginosa strains.  In conclusion, the goal should be a comparative analysis by including either OprF or mToxA to the observed protection.

Response:

To make the purpose and design of this experiment clear, we added the following sentences in the introduction and in the conclusion.

L64, “Considering that P. aeruginosa has various virulence factors and causes various infectious diseases, a multivalent vaccine targeting multiple antigens affecting major virulence factors is ideal for an anti-P. aeruginosa vaccine. For example, vaccines against pathogenic bacteria, such as pneumococcal vaccines, have been developed to be polyvalent vaccines effective against various pneumococcal subspecies [28]. Similarly, because P. aeruginosa has a variety of pathogenic mechanisms and causes many different types of infections, there has always been the question of whether a monovalent vaccine, such as the PcrV vaccine alone, would be able to cover the various aspects of P. aeruginosa infection. From this viewpoint, a trivalent P. aeruginosa vaccine was prepared and evaluated for protective efficacy against acute P. aeruginosa lung injury in this study.”

L431, “In this study, we created POmT as a single recombinant protein by combining PcrV, OprF, and mTox with glycine-serine linkers. Compared with vaccination with PcrV alone, immunization with POmT demonstrated comparable levels of all tested parameters, such as specific antibody titers, survival, lung edema, MPO, bacteriology in lungs, and lung histology. In addition, the protective effects of the POmT vaccine were almost the same as those of the mixture of PcrV, OprF, and mTox. In our series of mouse models, we were unable to demonstrate a benefit of including OprF and/or mTox as a vaccine component against P. aeruginosa pneumonia. We confirmed that immunity to PcrV is essential for preventing P. aeruginosa-induced acute lung injury and that immunity to OprF and exotoxin A has little effect. In addition, when using genetic recombination technology, even with the artificially synthesized trivalent vaccine, antibody titers were increased. Moreover, the impact on PcrV without affecting the antigenicity of components, including PcrV, was confirmed. Whether P. aeruginosa trivalent vaccine POmT is effective against other P. aeruginosa strains and different P. aeruginosa infections is an issue that will be addressed in future studies.”

2)      Figure 4-a: The survival rate among mice vaccinated with PMOT and PCRV was 79% and 78% respectively.  Is 78% significant but 78% not? (statistically speaking).

Response:

We corrected errors in the labeling of statistical significance in Figure 4. We re-checked the statistics, and the survival rates in the POmT, PcrV, and uninfected groups are significantly higher than in the alum only group. We corrected our explanation of these results in the Results section, as follows:

L256, “The 24-h survival rates were 79%, 78%, 21%, 7%, and 36% in the POmT, PcrV, OprF, mTox, and alum-alone groups, respectively (Figure 4a). Statistically, the POmT and PcrV group had significantly higher survival rates than the alum-alone group (p < 0.001 and p = 0.019, respectively). The survival rate in the mTox group was significantly lower than those in the alum alone (p = 0.012) and PcrV (p = 0.0002) groups.”

There were several mistakes in the labeling of statistical significance in Figures 4, 5, and 6. We rechecked the statistics of the multiple comparison survival curves, using post-hoc analysis for the log-rank test with p-value adjustment by the Benjamini & Hochberg method. Statistical analysis for the other graphs was also rechecked and corrected for Figures 4–6.

3)      Figure 4-a: It is intriguing that the survival rate among mice vaccinated with alum was higher than those vaccinated with either OprF or mToxA (36% vs. 21% and 7%).

Response:

As the reviewer pointed out, there were lower survival rates in the OprF alone, mTox-alone, and OprF + mTox groups than in the alum group. Although the exact reason remains unknown, it may be that OprF or mTox immunization resulted in adverse survival effects or it may reflect experimental variability. Nonetheless, our findings confirmed that OprF, mTox, and divalent OprF+mTox showed no positive immune effects. To explain this in the manuscript, the following text was added to the Discussion:

L399, "However, immunization with OprF alone, mTox alone, and OprF+mTox resulted in a lower survival rate than that in the alum group. It remained unknown as to whether immunization with OprF or mTox brought about an adverse immune effect on survival or whether the lower survival rate was due to the instability of the experiment. However, the OprF, mTox alone, and bivalent OprF+mTox vaccines did not demonstrate the positive immune effects seen with PcrV and POmT."

4)      Figure 4, a and b: It is intriguing that the survival rate of the combined vaccine (OprF + mToxA) is 9% whereas that of the single OprF is 21%. Does this suggest that mToxA lessen the efficacy of the OprF?

Response: As the reviewer pointed out, the survival rate of OprF alone was 21%, while the survival rate of OprF + mTox was 9%. In other words, the mTox component tends to decrease the survival rate. However, there was no statistically significant difference between the two groups (p = 0.73).

The reason for the negative effect of mTox on survival remains unknown. The text detailed above in response to question (3) was added to the Discussion to help explain this issue.

5)      Figure 4d: Survival analysis using PA103DpcrV: The authors compared the survival rate of mice vaccinated with either the trivalent vaccine or PCRV vaccine.  Since earlier experiments with the wt. PA103 (figure 4, a-c) clearly showed that PcrV is the most efficient, one would expect the authors to compare the groups vaccinated with OprF, mToxA, or a combination of those. PCRV vaccination against PA103Dpcrv will not lead to the required important comparison.  To delineate the contribution of OprF, mToxA, or both, the mutant lacks PcrV but still produces OprF and toxin A.

Response:

As the reviewer pointed out, PA103ΔpcrV lacks expression of PcrV, but the expression of OprF and exotoxin A is similar to that of wild-type PA103. PA103ΔpcrV is unable to exert type III secretory toxicity because of PcrV deficiency. Therefore, mortality following PA103ΔpcrV infection was not related to type III secretory toxicity. Because POmT contains mTox and OprF as components, we postulated that mortality rates due to PA103ΔpcrV infection may be improved compared with PcrV alone. However, PA103ΔpcrV was strongly attenuated because of a lack of type III secretory toxicity. The mortality rate of mice was about 50% even when a 100-times higher dose was administered. POmT resulted in a higher survival rate than PcrV alone, but the difference was not statistically significant. Therefore, we did not perform experiments with OprF or mTox alone. The interpretation of this result is described in the Discussion as follows:

L410,  ”No significant difference was observed between the POmT and PcrV vaccine groups. These findings indicate that the pathogenic mechanism of acute lung injury induced by P. aeruginosa is highly dependent on the type III secretion system and is largely unrelated to other pathogenic factors.”

6)      Possibly the authors need to provide a comparison of the survival rate among mice infected with either PA103 and PA103DpcrV (with or without vaccination).  This helps the readers appreciate the contribution of PcrV or the T3SS through PCRV.

Response:

As the reviewer suggested, we added the following sentences to the Results section:

L303, “The survival rates of PA103 (1×106 cfu)-infected mice treated with POmT, PcrV, and alum were 79%, 78%, and 36%, respectively, while the survival rates of PA103ΔpcrV (1×108 cfu)-infected mice treated with POmT, PcrV, and alum were 68%, 47%, and 50%, respectively. Although the bacterial dose of PA103ΔpcrV was 100 times higher than that of PA103, there was no statistically significant difference in mortality between PA103-infected and PA103ΔpcrV-infected mice and the corresponding vaccine groups.” 

7)      Figure 5-a, the lung edema index: The authors clearly showed that the index in mice vaccinated with POMT is significantly lower than those vaccinated with OprF-mToxA (53% vs. 66%).  However, the index in mice vaccinated with OprF or mToxA a is also low (55% and 58%).  The comparison should include all types of vaccinations.

Response:

We rechecked the statistics and confirmed that there was no statistically significant difference between POmT and OprF-mToxA. We added the following description to the text:

L313, “The lung edema index was 5.4 ±1.4 g in the POmT group , it was the highest in the OprF+mTox group (6.6 ± 0.8), although there was no statistically significant difference among the groups infected with PA103.”

8)      Figure 6, lung bacterial load: a) the authors stated that the lung bacterial counts were significantly lower in the POMT, PCVR, and PcrV + OprF + MToxA than mice treated with alum alone.  No significant deference was detected between mice treated with other groups and alum.  However, if one expects the observed significant reduction is due to the presence of PcrV, what about the combined treatment with PcrV + OprF or PcrV + mToxA? Both contain PCRV; b) the authors reported the lung bacterial load as CFU/lung.  However, variations likely to occur.  This should be standardized.  The reporting should be: CFU/gm of lung tissue.

Response:

Because all infected mice received the same bacterial dose (PA103 1x106 cfu/mouse, PA103ΔpcrV 1x108 cfu/mouse), calculating the total number of bacteria in the whole lung appears to be the easiest method to determine whether the bacterial population has increased or decreased in comparison with the number of instilled bacteria. The statistics also include the number of bacteria in mice after death, as displayed in closed circles; however, the actual number of bacteria is affected by the time of analysis after death, which complicates the interpretation of these results. We added the "instilled cfu-line" to the graph to enable the reader to understand and interpret this residual number of bacteria in the lung in relation to the number of bacteria administered.

We added an explanation to the legend of Figure 6 and to the Results section, as follows:

“Red dotted lines demonstrate the initial bacterial dose.”

L330, “More mice in the groups immunized with a vaccine lacking the PcrV component were found to have an increased number of bacteria in their lungs relative to the initial bacterial dose (§p < 0.05) (red dashed lines, Figures 6a and 6b).”.

9)      The backbone of this study is the trivalent vaccine (POmT).  Therefore, a possible section for the comparison of POMT with the PCRV + OPRF + MToxA vaccine is essential.  Is there a difference in any of the tested categories?

Response:

As the reviewer suggested, we added a description about the trivalent vaccine POmT to the Conclusions section:

L431, “In this study, we created POmT as a single recombinant protein by combining PcrV, OprF, and mTox with glycine-serine linkers. Compared with vaccination with PcrV alone, immunization with POmT demonstrated comparable levels of all tested parameters, such as specific antibody titers, survival, lung edema, MPO, bacteriology in lungs, and lung histology. In addition, the protective effects of the POmT vaccine were almost the same as those of the mixture of PcrV, OprF, and mTox. In our series of mouse models, we were unable to demonstrate a benefit of including OprF and/or mTox as a vaccine component against P. aeruginosa pneumonia. We confirmed that immunity to PcrV is essential for preventing P. aeruginosa-induced acute lung injury and that immunity to OprF and exotoxin A has little effect. In addition, when using genetic recombination technology, even with the artificially synthesized trivalent vaccine, antibody titers were increased. Moreover, the impact on PcrV without affecting the antigenicity of components, including PcrV, was confirmed. Whether P. aeruginosa trivalent vaccine POmT is effective against other P. aeruginosa strains and different P. aeruginosa infections is an issue that will be addressed in future studies.”

Reviewer 2 Report

The authors are trying to develop a trivalent vaccine including PcrV, OprF, and mTox. But its efficacy is not significant compared to single antigen PcrV and a simple mixture of three antigens in their animal model. although IgG titers against OprF, and mTox are increased after immunization with POmT, the single vaccination data shows that they are even worse than the alum control, so I am wondering if the format of antigen is effective in the tested model and no additive effect for granted. Or the animal model should be optimized. Unfortunately, I would not recommend it.

Minor comments:

1. line 292, is it correct? The data shows that the presence of PcrV is critical for preventing hypothermia in animals.

2. line 299, please delete "at 8"

3. line 389, I am confused with the "consist of a secretory apparatus... and a translocation mechanism.".

Author Response

Responses to the reviewer #2 comments

Reviewer #2

The authors are trying to develop a trivalent vaccine including PcrV, OprF, and mTox. But its efficacy is not significant compared to single antigen PcrV and a simple mixture of three antigens in their animal model. although IgG titers against OprF, and mTox are increased after immunization with POmT, the single vaccination data shows that they are even worse than the alum control, so I am wondering if the format of antigen is effective in the tested model and no additive effect for granted. Or the animal model should be optimized. Unfortunately, I would not recommend it.

Minor comments:

  1. line 292, is it correct? The data shows that the presence of PcrV is critical for preventing hypothermia in animals.

Response:

Thank you for pointing this out. We have correct this sentence and now state that the “presence” of PcrV is critical for preventing hypothermia in animals.  

  1. line 299, please delete "at 8"

Response:

We have deleted it.

  1. line 389, I am confused with the "consist of a secretory apparatus... and a translocation mechanism.".

Response:

We rewrote the sentences as follows:

L359, “The type III secretion system of gram-negative bacteria accomplishes the direct delivery of protein toxins from bacterium to host by nanosyringe “injectisomes,” which form a conduit across the two bacterial membranes, extracellular space, and the plasma membrane of a target eukaryotic cell. [15]. PcrV, which has a cap-like structure at the tip of the needle-like structure of the P. aeruginosa injectisome, is involved in the toxin translocation mechanism across the plasma membrane of a target eukaryotic cell. [15].”

Reviewer 3 Report

In this study Inoue et al. generate a new trivalent vaccine (POmT) and evaluate its immunogenicity and efficacy in a mouse model of acute lung infection caused by one strain of P. aeruginosa. For comparison, the authors included single-, two- and three-antigen mixed vaccines. A working vaccine against P. aeruginosa remains elusive being the goal of this manuscript of current interest. The authors show that POmT is immunogenic and protects mice from dead caused by PA103. POmT efficacy was not inferior to other vaccine formulations and authors claimed the importance of the PcrV antigen to be included in a vaccine against P. aeruginosa.

1) A significant limitation of the study is that the model is quite brief, running completion with just 24 hours of infection. The authors refer to a previous publication (Sawa et al. Nat Med 1999) in which most of mice infected with PA103 succumbed to infection before 24 hours. However, mice strains, vaccine formulations, immunization schedules and doses for challenging mice in both studies are not interchangeable. I suspect the outcome may be different if mice are monitored for longer term.

 2) Did the authors perform any a priori analysis to calculate sample size for animal studies? What was the reason for using different sample sizes for each group of mice? In particular, the group of mice immunized with POmT is up to three times larger than the groups for the remaining formulations? Could the different sample sizes imply a bias in the statistical analysis?

3) Based on recent work in the P. aeruginosa field, systemic humoral responses are unlikely to be solely responsible for protection against respiratory tract infections. Could the authors elaborate on the potential impact of SC vaccination on stimulation of mucosal immunity in the airway?

4) It is surprising that the ICR mice were 4 weeks old. However the majority of studies in the field perform vaccination in mice ranging 6-12 weeks old due to the mature of the immune system.

5) In view of Fig.4a the dose of PA103 for challenging mice was not the DL100. Notable the survival rate showed by the control group is substantially higher than rates observed in OprF and mTox vaccinated animals which seem to be inconsistent.

6) After challenging mice with PA103 lacking pcrV there is no significant vaccine protection with POmT and PcrV (?), and showed survival rates comparable to the saline control mice (Fig. 4d).

7) It would be useful for the reader if authors could explain Fig. 7 in more detail and point out in the microphotographs the foci of inflammatory infiltrates. How many mice per group were subjected to histological analysis? Are shown microphotographs representative?

8) I encourage authors to shorten the discussion and to focus on their results in relation to data from previous studies. As it is now I feel like it is more an introduction than a discussion.

Author Response

Responses to the reviewer #3 comments

Reviewer #3

In this study Inoue et al. generate a new trivalent vaccine (POmT) and evaluate its immunogenicity and efficacy in a mouse model of acute lung infection caused by one strain of P. aeruginosa. For comparison, the authors included single-, two- and three-antigen mixed vaccines. A working vaccine against P. aeruginosa remains elusive being the goal of this manuscript of current interest. The authors show that POmT is immunogenic and protects mice from dead caused by PA103. POmT efficacy was not inferior to other vaccine formulations and authors claimed the importance of the PcrV antigen to be included in a vaccine against P. aeruginosa.

 Response:

We appreciate your positive response and understanding of our experimental results.

  • A significant limitation of the study is that the model is quite brief, running completion with just 24 hours of infection. The authors refer to a previous publication (Sawa et al. Nat Med 1999) in which most of mice infected with PA103 succumbed to infection before 24 hours. However, mice strains, vaccine formulations, immunization schedules and doses for challenging mice in both studies are not interchangeable. I suspect the outcome may be different if mice are monitored for longer term.

Response:

We understand the reviewer's concerns about our experiment being terminated at 24 hours. Our animal model focuses on acute mortality and the treatment of infected individuals with acute P. aeruginosa lung injury depending on the P. aeruginosa type III secretion system. In fact, for mice that survived in this model at 24 hours, the initial acute lung injury was reduced, and effective prophylaxis (or treatment) was achieved. We do not suggest that infected mice will not die after 24 hours, but we believe that the purpose of our study is achieved in a 24-hour timescale. Therefore, we prioritized the infection experiment standards of our facility and considered the humane and ethical viewpoint regarding animal welfare. We added the following sentences to the Methods section and cited three additional references #35–#37 to help readers understand the model design:

L176, “As we reported previously, the mice which received a lethal dose of P. aeruginosa PA103 (1.0 × 106 cfu, LD50%~ 24 h) became hypothermic within 4 hours [36]. The hypothermic pathology and mortality within 24 h in this mouse model of pneumonia is not a result of bacterial multiplication in the infected organ but a consequence of cell necrosis of lung epithelial cells due to the translocation or the type III secretory toxin ExoU of P. aeruginosa into the lung epithelial cells within 4 h after infection [37,38]. We have also reported that the necrosis of lung epithelial cells due to ExoU toxin translocation occurs within 1 h after infection in an in-vitro culture cell model [36]. In addition, it has been reported that this pathology disappears after infection with an isogenic mutant strains lacking the enzymatically active ExoU of P. aeruginosa [36-38]. Therefore, our therapeutic model focuses on the first 4 to 24 h after infection. Accordingly, the experiment was completed in 24 h to focus on the pathogenesis of acute lung injury induced by P. aeruginosa, in addition to considerations on animal welfare and preventing contamination within the facility according to the infection experiment regulations of our facility.”

  1. Sawa, T., Corry, D.B., Gropper, M.A., Ohara, M., Kurahashi, K., Wiener-Kronish, J.P. IL-10 improves lung injury and survival in Pseudomonas aeruginosa pneumonia. J. Immunol., 1997, 159, 2858-2866.
  2. 36. Sawa, T., Ohara, M., Kurahashi, K., Twining, S.S., Frank, D.W., Doroques, D.B., Long, T., 35. Gropper, M.A., Wiener-Kronish, J.P. In vitro cellular toxicity predicts Pseudomonas aeruginosa virulence in lung infections. Immun. 1998, 66, 3242-3249. doi: 10.1128/IAI.66.7.3242-3249.1998.
  3. Pankhaniya, R.R., Tamura, M., Allmond, L.R., Moriyama, K., Ajayi, T., Wiener-Kronish, J.P., Sawa, T. Pseudomonas aeruginosa causes acute lung injury via the catalytic activity of the patatin-like phospholipase domain of ExoU. Crit. Care Med. 2004, 32, 2293-2299. doi:10.1097/01.ccm.0000145588.79063.07.
  4. Sato, H., Frank, D.W., Hillard, C.J., Feix, J.B., Pankhaniya, R.R., Moriyama, K., Finck-Barbancon, V., Buchaklian, A., Lei, M., Long, R.M., Wiener-Kronish, J., Sawa, T. 2003. The mechanism of action of the Pseudomonas aeruginosa-encoded type III cytotoxin, ExoU. EMBO J. 2003, 22, 2959-2969.

  • Did the authors perform any a priori analysis to calculate sample size for animal studies? What was the reason for using different sample sizes for each group of mice? In particular, the group of mice immunized with POmT is up to three times larger than the groups for the remaining formulations? Could the different sample sizes imply a bias in the statistical analysis?

Response:

The time taken to handle the experimental animals (manipulations such as general anesthesia and bacterial infection, removal of organs after euthanasia, and their preservation) meant that in our laboratory, the number of mice that could be handled in one experiment was limited to 14–15. In addition, depending on how the number of bacteria is adjusted, the survival rates of animals may vary. To ensure consistency, we routinely included a positive control group (POmT) and a negative control group (alum alone), and along with the seven test groups (PcrV, OprF, mTox, PcrV+OprF, PcrV+mTox, OprF+mTox, PcrV+OprF+mTox), all nine groups were tested at least three times to confirm reproducibility. Experiments generally involved at least 12 animals and were repeated several times on different days. For example, in one experiment, 14 animals were included: four animals in the POmT group, four animals in the PcrV group, four animals in the OprF group, and two animals in the alum group. In another experiment, the 14 animals included four in the POmT, four in the OprF group, four in the mTox group, and two in the alum group.

Therefore, in total, 42 animals were tested in the POmT positive control group, and 28 animals were tested in the alum negative control group. An explanation has been added to the Methods section as follows:

L189, "In a single experiment, an experiment was planned with a maximum of 15 mice, including the POmT group as a positive control (effective) group and the alum group as a negative control (ineffective) group, with other groups to be investigated interposed in between to determine the effect. Experiments were repeated at least three times to confirm reproducibility. More than 10 animals/group were obtained with a minimum of three experiments to ensure reproducibility. For this reason, the PA103 infection series has 42 animals in the POmT group and 28 animals in the alum group, which is larger than the other groups (10-18 animals)."

  • Based on recent work in the  aeruginosafield, systemic humoral responses are unlikely to be solely responsible for protection against respiratory tract infections. Could the authors elaborate on the potential impact of SC vaccination on stimulation of mucosal immunity in the airway?

Response:

As the reviewer pointed out, airway mucosal immunity is crucial in the prevention of respiratory tract infections and is affected by the vaccine administration method. We recently reported the efficacy of PcrV nasal vaccine administration, including an increase in specific IgA secretion in the airways (ref #12). Therefore, we also recognize the importance of intranasal administration for this POmT vaccine. However, in this study, we focused on comparing POmT and PcrV, and chose subcutaneous administration as a more conventional established approach. Regarding the efficacy of intranasal administration of POmT, this would be an area of interest in future studies and we have added this explanation to the Discussion section as follows:

L423, "Finally, in this experiment, all prepared component vaccines were administered subcutaneously to evaluate the protective effect of the POmT vaccine with those administered using more conventional immunization methods. P. aeruginosa is a pathogen that infects the respiratory tract, so airway mucosal immunity is essential. As we recently reported the efficacy of intranasal administration of the PcrV-CpG deoxyoligonucleotide vaccine [12], our future studies will focus on confirming the effectiveness of intranasal administration of the POmT vaccine."

  • It is surprising that the ICR mice were 4 weeks old. However the majority of studies in the field perform vaccination in mice ranging 6-12 weeks old due to the mature of the immune system.

Response:

In our vaccination experiments, 4-week-old ICR mice were first immunized, then 8 weeks later, infection experiments were performed. In all other infection experiments, 12-week-old mice were used. The reason for this is that after 12 weeks, the size of the mice makes it technically difficult to conduct infection experiments by tracheal administration under general anesthesia, and the number of bacteria instilled and the infection status become more variable. For this reason, we started vaccination at 4 weeks of age. Using a blood sample at 11 weeks of age, IgG was measured as confirmation of immune function, and no significant increase was observed.

The information that the infection experiments were performed at 12 weeks of age was added to the Methods section.

  • In view of Fig.4a the dose of PA103 for challenging mice was not the DL100. Notable the survival rate showed by the control group is substantially higher than rates observed in OprF and mTox vaccinated animals which seem to be inconsistent.

Response:

As noted by the reviewers, the dose of P. aeruginosa was actually LD67% at 24 hours. Ideally, LD50% is more effective for judging efficacy and side effects. In experiments using P. aeruginosa strain PA103, it was necessary to dilute the concentration of the bacteria from 109 cfu/mL, which can be measured by spectrometry, to 107 cfu/mL, a 100-fold dilution. Because of technical limitations, a variation of approximately ±15% was recognized depending on the experimental results, even if the target was LD50%. For this reason, as described above, experiments were repeated at least three times for each group and included a positive control group and a negative control group. As the reviewer pointed out, OprF alone, mTox alone, and OprF + mTox resulted in a lower survival rate than the alum group. It remains unknown whether OprF or mTox immunization resulted in adverse survival effects or whether it was due to experimental variability. Nonetheless, OprF, mTox, and divalent OprF+mTox showed no positive immune effects. To explain this, the following text was added to the Discussion section:

L399, "However, immunization with OprF alone, mTox alone, and OprF+mTox resulted in a lower survival rate than that in the alum group. It remained unknown as to whether immunization with OprF or mTox brought about an adverse immune effect on survival or whether the lower survival rate was due to the instability of the experiment. However, the OprF, mTox alone, and bivalent OprF+mTox vaccines did not demonstrate the positive immune effects seen with PcrV and POmT."

  • After challenging mice with PA103 lacking pcrV there is no significant vaccine protection with POmT and PcrV (?), and showed survival rates comparable to the saline control mice (Fig. 4d).

Response:

Thank you, we have corrected Figure 4d. There was no statistically significant difference between the survival of the uninfected and PcrV-vaccinated groups. We also added the relevant explanation to the Results section, as follows:

L299, “The survival rate of PcrV-vaccinated mice infected with PA103ΔpcrV was 47%, which was not significantly lower than that of uninfected mice (p = 0.23), while the survival rates in the alum and POmT groups were 50% and 68%, respectively. Although the survival rate with POmT was higher than that with PcrV, this difference was not statistically significant (p = 0.23).”

  • It would be useful for the reader if authors could explain Fig. 7 in more detail and point out in the microphotographs the foci of inflammatory infiltrates. How many mice per group were subjected to histological analysis? Are shown microphotographs representative? 

Response:

Lung tissue was collected from surviving mice (two mice per group) 24 hours after administration of P. aeruginosa, and was subjected to histological evaluation. Representative images of the lung tissue of one mouse from the two mice analyzed, are presented in the figure. Because the lung tissue was collected from mice that had survived for 24 hours, it is assumed that the inflammation is less severe than the average for the group. This information was added to the Methods section and to the legend of Figure 7:

L337, “By contrast, the lungs of POmT-vaccinated mice and PcrV-vaccinated mice displayed significantly fewer inflammatory changes than those of the other groups. It is assumed that lung histology showed less inflammation than the average for the group because lung tissue was collected from surviving mice over 24 h in each group.”

  • I encourage authors to shorten the discussion and to focus on their results in relation to data from previous studies. As it is now I feel like it is more an introduction than a discussion.

Response:

As suggested by the reviewer, we have deleted most of the descriptions of past P. aeruginosa vaccine reports, other than those concerning OprF and exotoxin A, from the Discussion for simplification, as follows:

Pseudogen, which contained the lipopolysaccharide (LPS) O antigen of seven different serotypes, advanced to early-phase clinical trials [38,39]. However, the clinical effect was poor, and adverse events were identified [40,41]. Flagella (flagellin and flagella) are believed to be involved in the invasion of P. aeruginosa into the host, and they are involved in the activation of TLR5. The phase III clinical trial of a conjugate vaccine of flagellin coupled to polymannuronic acid was conducted in the 2000s in patients with cystic fibrosis [42]. However, this vaccine candidate only prevented 34% of acute infections. In addition, because it was ineffective against different P. aeruginosa flagella serotype, this vaccine antigen was not further investigated.

…..

Saha et al. focused on the pathogenicities of the outer membrane protein OprF/I, type III secretion system PcrV, and fimbriae PilA and verified the therapeutic effect of a multivalent DNA vaccine that mixed these multiple antigens [55]. The researchers compared the trivalent vaccine with each monovalent vaccine in a model of P. aeruginosa pneumonia and found that 10-day survival was better for the trivalent vaccine than for the monovalent vaccines. Jiang et al. developed a bivalent vaccine that genetically combined the extracellular protein toxins exotoxin A and PcrV [56]. They reported that the reduction of inflammatory tissue changes was significant with the multivalent vaccine. Yang et al. investigated the preventive effect of a vaccine genetically fused with three P. aeruginosa antigens, including PcrV, using mouse pneumonia and burn models [54]. They demonstrated that in mice with P. aeruginosa (PAO1) pneumonia, the trivalent vaccine PcrV-OprI-Hcp1 (a central component of the Hcp1:6-type secretion system involved in direct delivery of toxin proteins into infected host cells) was linked to a significantly higher survival rate than each monovalent vaccine.

Reviewer 4 Report

The manuscript titled "Effect of a novel trivalent vaccine formulation against acute lung injury caused by Pseudomonas aeruginosa" (vaccines-2405212) presents a comparative study of the POmT vaccine with three monovalent antigens (PcrV, OprF, and mTox) and two- and three-antigen combinations using a mouse model of P. aeruginosa pneumonia. The text is well-written, but some minor aspects need improvement.

Specific comments:

Page 3, line 122: Is there a specific reason for using young ICR mice (only 4 weeks old) for immunizations?

Page 4, Table 1: Why is the sample size (n) so different between vaccination groups? What is the reason for the sample size of n=42 in the POmT group?

Page 5, line 179: Reference 34 is not adequate for the text.

Page 5, lines 185-188: The authors state that their previous experiments (of reference 13) justify the observation period of only 24 hours after mice infection; however, the acute infection model used in both studies is different, including the mouse strains, the age of mice, the volume instilled in the lungs, the bacterial dose, and the vaccines used for immunizations. The authors should provide an adequate explanation.

Page 6, lines 241-242: The sentence in lines 241-242 means the same as the previous sentence (lines 238-241). Please remove one of them.

Page 7, Figure 3b and paragraph related (lines 245-248) are not relevant. Antibody titers are induced by vaccination, independent of the P. aeruginosa strain used for infection. However, it would be interesting to compare the level of specific IgGs against POmT and the combined two- and three-antigens vaccines in the vaccinated group of mice.

Page 8, Figure 4: Is there any significant difference between the POmT and PcrV groups of mice with respect to 24-h survival rate and body temperature in surviving mice 12 h after infection (Figure 4, panel A)?

Page 9, subsection 3.2.4., and lines 311-312: The result regarding the protective efficacy of the POmT and PcrV vaccines on mice infected with the PA103ΔpcrV strain (Figure 4d) is missing in the text.

Page 9, lines 312-314: The phrase has already been discussed in subsection 3.1.

Page 9, lines 332-334: The text does not clearly describe Figure 6b, perhaps due to the high intra-group variability (CFUs in lung) and the difference in sample size between the control and vaccinated groups. Please provide justification for the sentence.

Minor comments:

Please check that P. aeruginosa (lines 35, 39, 41, 43, etc.) and Escherichia coli (lines 87, 117) are written in italics throughout the text.

Page 9, line 331: Please change "(Figure 6 left)" to "(Figure 6a)".

Page 9, line 334: Please change "(Figure 6 right)" to "(Figure 6b)".

Page 10, Figure 6 legend: Please indicate panels a) and b).

Page 10, Figure 6: The "dotted lines in the box" indicated in the legend are missing in the figure.

Author Response

Responses to the reviewer #4 comments

Reviewer #4

The manuscript titled "Effect of a novel trivalent vaccine formulation against acute lung injury caused by Pseudomonas aeruginosa" (vaccines-2405212) presents a comparative study of the POmT vaccine with three monovalent antigens (PcrV, OprF, and mTox) and two- and three-antigen combinations using a mouse model of P. aeruginosa pneumonia. The text is well-written, but some minor aspects need improvement.

Specific comments:

Page 3, line 122: Is there a specific reason for using young ICR mice (only 4 weeks old) for immunizations?

Response:

In the vaccination experiments, 4-week-old ICR mice were first immunized, and 8 weeks later, infection experiments were performed. In other infection experiments, 12-week-old mice were used. The reason for this is that the size of the mice at 12 weeks makes it technically difficult to conduct infection experiments by tracheal administration under general anesthesia, and the number of bacteria instilled and the infection status become more variable. For this reason, we started vaccination at 4 weeks of age. Using a blood sample at 11 weeks of age, IgG was measured as confirmation of immune function, and no significant increase was observed.

It was added to the Methods section that the infection experiment was performed at 12 weeks of age.

Page 4, Table 1: Why is the sample size (n) so different between vaccination groups? What is the reason for the sample size of n=42 in the POmT group?

Response:

The time taken to handle the experimental animals (manipulations such as general anesthesia and bacterial infection, removal of organs after euthanasia, and their preservation) meant that in our laboratory, the number of mice that could be handled in one experiment was limited to 14–15. In addition, depending on how the number of bacteria is adjusted, the survival rates of animals may vary. To ensure consistency, we routinely included a positive control group (POmT) and a negative control group (alum alone), and along with the seven test groups (PcrV, OprF, mTox, PcrV+OprF, PcrV+mTox, OprF+mTox, PcrV+OprF+mTox), all nine groups were tested at least three times to confirm reproducibility. Experiments generally involved at least 12 animals and were repeated several times on different days. For example, in one experiment, 14 animals were included: four animals in the POmT group, four animals in the PcrV group, four animals in the OprF group, and two animals in the alum group. In another experiment, the 14 animals included four in the POmT, four in the OprF group, four in the mTox group, and two in the alum group.

Therefore, in total, 42 animals were tested in the POmT positive control group, and 28 animals were tested in the alum negative control group. An explanation has been added to the Methods section as follows:

L189, "In a single experiment, an experiment was planned with a maximum of 15 mice, including the POmT group as a positive control (effective) group and the alum group as a negative control (ineffective) group, with other groups to be investigated interposed in between to determine the effect. Experiments were repeated at least three times to confirm reproducibility. More than 10 animals/group were obtained with a minimum of three experiments to ensure reproducibility. For this reason, the PA103 infection series has 42 animals in the POmT group and 28 animals in the alum group, which is larger than the other groups (10-18 animals)."

Page 5, line 179: Reference 34 is not adequate for the text.

Response:

We have replaced this reference with a more appropriate one:

Sawa, T., Corry, D.B., Gropper, M.A., Ohara, M., Kurahashi, K., Wiener-Kronish, J.P. IL-10 improves lung injury and survival in Pseudomonas aeruginosapneumonia. J. Immunol., 1997, 159, 2858-2866.

Page 5, lines 185-188: The authors state that their previous experiments (of reference 13) justify the observation period of only 24 hours after mice infection; however, the acute infection model used in both studies is different, including the mouse strains, the age of mice, the volume instilled in the lungs, the bacterial dose, and the vaccines used for immunizations. The authors should provide an adequate explanation.

Response:

We understand the reviewer's concerns about our experiment being terminated at 24 hours. Our animal model focuses on acute mortality and the treatment of infected individuals with acute P. aeruginosa lung injury depending on the P. aeruginosa type III secretion system. In fact, for mice that survived in this model at 24 hours, the initial acute lung injury was reduced, and effective prophylaxis (or treatment) was achieved. We do not suggest that infected mice will not die after 24 hours, but we believe that the purpose of our study is achieved in a 24-hour timescale. Therefore, we prioritized the infection experiment standards of our facility and considered the humane and ethical viewpoint regarding animal welfare. We added the following sentences to the Methods section and cited three additional references #35–#37 to help readers understand the model design:

L176, “As we reported previously, the mice which received a lethal dose of P. aeruginosa PA103 (1.0 × 106 cfu, LD50%~ 24 h) became hypothermic within 4 hours [36]. The hypothermic pathology and mortality within 24 h in this mouse model of pneumonia is not a result of bacterial multiplication in the infected organ but a consequence of cell necrosis of lung epithelial cells due to the translocation or the type III secretory toxin ExoU of P. aeruginosa into the lung epithelial cells within 4 h after infection [37,38]. We have also reported that the necrosis of lung epithelial cells due to ExoU toxin translocation occurs within 1 h after infection in an in-vitro culture cell model [36]. In addition, it has been reported that this pathology disappears after infection with an isogenic mutant strains lacking the enzymatically active ExoU of P. aeruginosa [36-38]. Therefore, our therapeutic model focuses on the first 4 to 24 h after infection. Accordingly, the experiment was completed in 24 h to focus on the pathogenesis of acute lung injury induced by P. aeruginosa, in addition to considerations on animal welfare and preventing contamination within the facility according to the infection experiment regulations of our facility.”

  1. Sawa, T., Corry, D.B., Gropper, M.A., Ohara, M., Kurahashi, K., Wiener-Kronish, J.P. IL-10 improves lung injury and survival in Pseudomonas aeruginosa pneumonia. J. Immunol., 1997, 159, 2858-2866.
  2. 36. Sawa, T., Ohara, M., Kurahashi, K., Twining, S.S., Frank, D.W., Doroques, D.B., Long, T., 35. Gropper, M.A., Wiener-Kronish, J.P. In vitro cellular toxicity predicts Pseudomonas aeruginosa virulence in lung infections. Immun. 1998, 66, 3242-3249. doi: 10.1128/IAI.66.7.3242-3249.1998.
  3. Pankhaniya, R.R., Tamura, M., Allmond, L.R., Moriyama, K., Ajayi, T., Wiener-Kronish, J.P., Sawa, T. Pseudomonas aeruginosa causes acute lung injury via the catalytic activity of the patatin-like phospholipase domain of ExoU. Crit. Care Med. 2004, 32, 2293-2299. doi:10.1097/01.ccm.0000145588.79063.07.
  4. Sato, H., Frank, D.W., Hillard, C.J., Feix, J.B., Pankhaniya, R.R., Moriyama, K., Finck-Barbancon, V., Buchaklian, A., Lei, M., Long, R.M., Wiener-Kronish, J., Sawa, T. 2003. The mechanism of action of the Pseudomonas aeruginosa-encoded type III cytotoxin, ExoU. EMBO J. 2003, 22, 2959-2969.

Page 6, lines 241-242: The sentence in lines 241-242 means the same as the previous sentence (lines 238-241). Please remove one of them.

Response:

We have deleted this redundant sentence, as suggested.

Page 7, Figure 3b and paragraph related (lines 245-248) are not relevant. Antibody titers are induced by vaccination, independent of the P. aeruginosa strain used for infection. However, it would be interesting to compare the level of specific IgGs against POmT and the combined two- and three-antigens vaccines in the vaccinated group of mice.

Response:

As the reviewer pointed out, specific antibody titers against PcrV, OprF, and mTox are unrelated to the bacterial strain used for infection. However, we kept Figure 3b to show successful vaccination in the PA103ΔpcrV-infection group. In this study, we did not measure the increase in specific antibody titers with bivalent and trivalent vaccines because vaccination with a single antigen and POmT significantly and reliably increased the specific titers.

Page 8, Figure 4: Is there any significant difference between the POmT and PcrV groups of mice with respect to 24-h survival rate and body temperature in surviving mice 12 h after infection (Figure 4, panel A)?

Response:

There was no statistically significant difference in the survival rate or body temperatures between mice in the POmT and PcrV groups. This information has been added to the Results section:

L265, “There was no statistically significant difference in the survivals and the body temperatures hours between POmT and PcrV.”

Page 9, subsection 3.2.4., and lines 311-312: The result regarding the protective efficacy of the POmT and PcrV vaccines on mice infected with the PA103ΔpcrV strain (Figure 4d) is missing in the text.

Response:

We hesitate to state the protective effect of POmT and PcrV in mice infected with PA103ΔpcrV strain because there was no statistically significant difference in the survival rates between the POmT, PcrV, and alum groups, although there was a higher survival rate with POmT than with alum or PcrV. We added the following sentences to the Results section to explain this:

L299, “The survival rate of PcrV-vaccinated mice infected with PA103ΔpcrV was 47%, which was not significantly lower than that of uninfected mice (p = 0.23), while the survival rates in the alum and POmT groups were 50% and 68%, respectively. Although the survival rate with POmT was higher than that with PcrV, this difference was not statistically significant (p = 0.23). The survival rates of PA103 (1×106 cfu)-infected mice treated with POmT, PcrV, and alum were 79%, 78%, and 36%, respectively, while the survival rates of PA103ΔpcrV (1×108 cfu)-infected mice treated with POmT, PcrV, and alum were 68%, 47%, and 50%, respectively. Although the bacterial dose of PA103ΔpcrV was 100 times higher than that of PA103, there was no statistically significant difference in mortality between PA103-infected and PA103ΔpcrV-infected mice and the corresponding vaccine groups.”

Page 9, lines 312-314: The phrase has already been discussed in subsection 3.1.

Response:

We deleted this sentence.

Page 9, lines 332-334: The text does not clearly describe Figure 6b, perhaps due to the high intra-group variability (CFUs in lung) and the difference in sample size between the control and vaccinated groups. Please provide justification for the sentence.

Response:

Because all infected mice received the same bacterial dose (PA103 1x106 cfu/mouse, PA103ΔpcrV 1x108 cfu/mouse), calculating the total number of bacteria in the whole lung appears to be the easiest method to determine whether the bacterial population has increased or decreased in comparison with the number of instilled bacteria. The statistics also include the number of bacteria in mice after death, as displayed in closed circles; however, the actual number of bacteria is affected by the time of analysis after death, which complicates the interpretation of these results. We added the "instilled cfu-line" to the graph to enable the reader to understand and interpret this residual number of bacteria in the lung in relation to the number of bacteria administered.

We added an explanation to the legend of Figure 6 and to the Results section, as follows:

“Red dotted lines demonstrate the initial bacterial dose.”

L330, “More mice in the groups immunized with a vaccine lacking the PcrV component were found to have an increased number of bacteria in their lungs relative to the initial bacterial dose (§p < 0.05) (red dashed lines, Figures 6a and 6b).”.

Minor comments:

Please check that P. aeruginosa (lines 35, 39, 41, 43, etc.) and Escherichia coli (lines 87, 117) are written in italics throughout the text.

Response:

We have corrected this.

Page 9, line 331: Please change "(Figure 6 left)" to "(Figure 6a)".

Response:

We have made this change.

Page 9, line 334: Please change "(Figure 6 right)" to "(Figure 6b)".

Response:

We have made this change.

Page 10, Figure 6 legend: Please indicate panels a) and b).

Response:

We have made this change.

Page 10, Figure 6: The "dotted lines in the box" indicated in the legend are missing in the figure.

Response:

We have corrected this.

Round 2

Reviewer 1 Report

All my points are addressed.  Thank you.  I am still not convinced with the author’s response regarding the bacterial load within the lung.  The weight of the lung will likely varies and it is safer to express the findings as CFU/gm of tissues.

Author Response

Reviewer #1 All my points are addressed. Thank you. I am still not convinced with the author’s response regarding the bacterial load within the lung. The weight of the lung will likely varies and it is safer to express the findings as CFU/gm of tissues.

Response: As the reviewer suggested, we fixed the unit of Figure 6 for bacteria in lung to cfu/gram of lung tissue. The statistics were all recalculated for them. The following changes were made for the revision.

L218, The sequentially diluted solution of the lung homogenate was inoculated on a sheep blood agar plate and incubated at 37°C overnight. The number of cfu on each plate was used to determine the number of bacteria that existed in 100 µL of the homogenate, and calculate the total number of remaining bacteria in gram of lung tissues.

L329, Total bacterial counts in the lungs were calculated in both surviving and dead mice 24 h after P. aeruginosa infection (Figure 6). Following PA103 infection, lung bacterial counts were significantly lower in the PcrV+OprF+mTox group than in the alum-alone group (Figure 6a) (†p = 0.006). No significant differences were observed between the other groups and the alum-alone group. Following PA103ΔpcrV infection, the number of bacteria was lower in the POmT group than in the PcrV and alum groups, albeit without statistical significance (Figure 6b). More mice in the group immunized with POmT were found to have a decreased number of bacteria in their lungs relative to the initial bacterial dose (cfu/gram of lung tissue) (§p < 0.05) (red dashed lines, Figures 6a).

Figure 6 legend, Figure 6. Bacterial colony-forming units (cfu) in lung homogenates from the vaccinated mice infected with P. aeruginosa. Bacterial cfu in mouse lungs 24 h after intratracheal infection with P. aeruginosa. a) PA103-infected groups. b) PA103ΔpcrV-infected groups. Data (cfu/gram of lung tissue) are presented as the median (a center solid bar in the box) and the first (dotted lines in the box) and 5th & 95th percentiles (boxes) with each value indicated by symbols (open symbol: survived, closed symbol: died). †p < 0.05 compared with the alum-alone-vaccinated group. Proportion (%): the percentage of mice in which the number of bacteria remaining in the lung was ≥ the initial bacterial dose. §p < 0.05 for the rate (%) of occurrence of a greater number of remaining bacteria in the lung than the initial bacterial dose (5.2 × 106 cfu/gram infection and 5.2 × 108 cfu/gram of normal lung tissue for PA103 and PA103ΔpcrV-infection, respectively). Red dashed lines demonstrate the initial bacterial dose. PcrV, full-length PcrV; OprF, the outer membrane domain (#190–342) of OprF; mTOX, a non-catalytic mutated domain (#406–613, E553Δ) of exotoxin A; POmT, a conjugate of PcrV, and parts of OprF and mutated exotoxin A.

Reviewer 2 Report

The authors seemed not to give any responses to my major concern. 

Author Response

(The 1st comments): The authors are trying to develop a trivalent vaccine including PcrV, OprF, and mTox. But its efficacy is not significant compared to single antigen PcrV and a simple mixture of three antigens in their animal model. although IgG titers against OprF, and mTox are increased after immunization with POmT, the single vaccination data shows that they are even worse than the alum control, so I am wondering if the format of antigen is effective in the tested model and no additive effect for granted. Or the animal model should be optimized. Unfortunately, I would not recommend it.

(The 2nd comments): The authors seemed not to give any responses to my major concern.

Response:

Since our discovery in 1999 of the immune effect of the PcrV antigen against P. aeruginosa pneumonia, we have been working on developing PcrV-targeted therapeutic antibodies and PcrV vaccines. However, since P. aeruginosa retains diverse pathogenic mechanisms and causes various infectious diseases, many pharmaceutical companies have discouraged the development of immunotherapies targeting only a single PcrV antigen, mainly because of its potential narrow spectrum for pathogenicity. No vaccine against P. aeruginosa has been clinically applied to date. Since there is no evidence that a multi-antigen vaccine is superior to a PcrV single-antigen vaccine in P. aeruginosa infection, we investigated this by comparing vaccines targeting only PcrV with vaccines that include other antigens. Unfortunately, as described above, POmT, a trivalent vaccine, failed to show a definite advantage over the mono-antigen PcrV vaccine. However, the findings of this experiment revealed several important points. First, we confirmed that P. aeruginosa type III secretory toxicity and its inhibition are critical factors in the etiology and prevention of P. aeruginosa -induced acute lung injury, with other virulence factors playing a minor role. Second, the three antigens, which were artificially linked via an amino-acid linker in the molecular structure, retained the antigenicity and immune effects of PcrV. This finding was significant because, in producing vaccines containing three separate components, manufacturing regulations require that safety and efficacy tests are conducted for each element, including antigens and adjuvants, complicating the development of multi- antigen vaccines and raising costs considerably. Therefore, combining antigens at the gene level to generate a single molecule is considered the most realistic option when manufacturing a multi-protein antigen vaccine. As mentioned above, there is a concern that vaccines that do not contain PcrV (OprF and/or mTox), at least in our P. aeruginosa pneumonia model, may exacerbate rather than improve lung injury. In case of immunotherapies against virus infections, the possibility that immunization against specific antigens that do not suppress toxicity exacerbates cellular damage, especially of phagocytic cells, has been reported as the antibody-dependent enhancement (ADE) phenomenon. This mechanism of ADE involves the interaction of pathogen–antibody immune complexes to phagocytic cells through the adherence of the antibody Fc region with cellular Fc receptors. In this phenomenon, non-neutralizing antibodies generated by vaccination and bound to pathogen surface proteins, can promote the subsequent contact of pathogens to the target cells and intensify the inflammatory process during infection. Therefore, antibodies derived from vaccines that do not suppress cytotoxicity may promote the interaction of pathogens to phagocytic cells, and eventually induce more severe cytotoxicity. A mechanism similar to the ADE phenomenon may help to explain the poor performance of the present PcrV-free OprF and/or mTox vaccine.

We added the following paragraph explaining the above viewpoint, which is associated with the reviewer's concern about our experimental results. We cited two more references (46, 47) for the above explanation.

L737 of the tracked version, Since our discovery in 1999 of the immune effect of the PcrV antigen against P. aeruginosa pneumonia [8], we have been working on developing PcrV-targeted therapeutic antibodies and PcrV vaccines. However, since P. aeruginosa retains diverse pathogenic mechanisms and causes various infectious diseases [5], many pharmaceutical companies have discouraged the development of immunotherapies targeting only a single PcrV antigen, mainly because of its potential narrow spectrum against various P. aeruginosa pathogenicities. No vaccine against P. aeruginosa has been clinically applied to date [4]. Since there is no evidence that a multi-antigen vaccine is superior to a PcrV single-antigen vaccine in P. aeruginosa infection, we investigated this by comparing vaccines targeting only PcrV with vaccines that include other antigens. Unfortunately, as described above, POmT, a trivalent vaccine, failed to show a definite advantage over the mono-antigen PcrV vaccine. However, the findings of this experiment revealed several important points. First, we confirmed that P. aeruginosa type III secretory toxicity and its inhibition are critical factors in the etiology and prevention of P. aeruginosa-induced acute lung injury, with other virulence factors playing a minor role. Second, the three antigens, which were artificially linked via an amino-acid linker in the molecular structure, retained the antigenicity and immune effects of PcrV. This finding was significant because, in producing vaccines containing three separate components, manufacturing regulations require that safety and efficacy tests are conducted for each element, including antigens and adjuvants [42], complicating the development of multi-antigen vaccines and raising costs considerably. Therefore, combining antigens at the gene level to generate a single molecule is considered the most realistic option when manufacturing a multi-protein antigen vaccine. As mentioned above, there is a concern that vaccines that do not contain PcrV (OprF and/or mTox), at least in our P. aeruginosa pneumonia model, may exacerbate rather than improve lung injury. In the case of immunotherapies against virus infections, the possibility that immunization against specific antigens that do not suppress toxicity exacerbates cellular damage, especially of phagocytic cells, has been reported as the antibody-dependent enhancement (ADE) phenomenon [43]. This mechanism of ADE involves the interaction of pathogen–antibody immune complexes to phagocytic cells through the adherence of the antibody Fc region with cellular Fc receptors. In this phenomenon, non-neutralizing antibodies generated by vaccination and bound to pathogen proteins can promote the subsequent contact of pathogens to the target cells and intensify the inflammatory process during infection. Therefore, antibodies derived from vaccines that do not suppress cytotoxicity may promote the interaction of pathogens with phagocytic cells and eventually induce more severe cytotoxicity. A mechanism similar to the ADE phenomenon may help to explain the poor performance of the present PcrV-free OprF and/or mTox vaccine.

Reviewer 3 Report

The authors have responded to my comments and I appreciate their effort to improve the manuscript. Please, see below a couple of minor points, which I think should be adressed before acceptance of the paper.

If I have understood correctly the authors’ response, the objective of the study is to evaluate the vaccine efficacy in the first 24 h after infection, and then the parameters indicating acute lung injury (hypotermia, edema index, MPO activity…) were considered more relevant that others (lung bacterial loads, survival..) in this model. If this is indeed correct, it must be clearly stated in the manuscript and I strongly suggest the authors to included results derived from these experiments in the abstract.

I thank authors for the explanation about sample size (lines 189-196). However, using different size for mice groups could substantially affect statistics. I recommend authors to consider a priori sample calculation for future studies (see a recent commentary on this in PMID: 33380887).

I think it is confusing for the reader the term “therapeutic” when speaking for vaccines. I consider a therapeutic approach when it is administered to an already infected individual with the objective to treat an infection. In this work all vaccine candidates are injected prophylactically before infection so I strongly recommend to amend the text accordingly.

Author Response

The authors have responded to my comments and I appreciate their effort to improve the manuscript. Please, see below a couple of minor points, which I think should be addressed before acceptance of the paper.

If I have understood correctly the authors’ response, the objective of the study is to evaluate the vaccine efficacy in the first 24 h after infection, and then the parameters indicating acute lung injury (hypothermia, edema index, MPO activity…) were considered more relevant that others (lung bacterial loads, survival..) in this model. If this is indeed correct, it must be clearly stated in the manuscript and I strongly suggest the authors to included results derived from these experiments in the abstract.

Response: We revised the abstract, the introduction, and the conclusion, to put emphasis on the purpose of this vaccine as prophylaxis for acute lung injury.

Abstract, An effective vaccine against Pseudomonas aeruginosa would be beneficial for people susceptible to severe infection. Vaccination targeting V antigen (PcrV) of the P. aeruginosa type III secretion system is a potential prophylactic strategy for reducing P. aeruginosa-induced acute lung injury and the acute mortality. We created a recombinant protein (designated POmT) comprising three antigens: full-length PcrV (PcrV#1-#294), the outer membrane domain (#190–342) of OprF (OprF#190-#342), and a non-catalytic mutant of the carboxyl domain (#406–613) of exotoxin A (mToxA#406-#613(E553Δ)). In the combination of PcrV and OprF, mToxA, the efficacy of POmT was compared with that of single-antigen vaccines, two-antigen mixed vaccines, and a three-antigen mixed vaccine in a murine model of P. aeruginosa pneumonia. As a result, the 24 h-survival rates were 79%, 78%, 21%, 7%, and 36% in the POmT, PcrV, OprF, mTox, and alum-alone groups, respectively. Significant improvement in acute lung injury and reduction in acute 

mortality within 24 hours after infection was observed in the POmT and PcrV groups than in the other groups. Overall, the POmT vaccine exhibited efficacy comparable to that of the PcrV vaccine. The future goal is to prove the efficacy of the POmT vaccine against various P. aeruginosa strains.

L91 of the tracked version, Our past animal studies have shown that active immunization with recombinant PcrV can significantly reduce acute pulmonary epithelial injury that occurs and exacerbates within 4 h following a lethal dose of P. aeruginosa pulmonary infection. Consequently, active immunization with PcrV can suppress subsequent pulmonary edema and inflammation, improve bacterial clearance in the lung, and reduce acute mortality within 24 h. Therefore, using our murine model in the current study, we evaluated the preventive effect of the newly created three-antigen POmT against P. aeruginosa-induced acute lung injury by comparing the prophylactic effects against recombinant PcrV and two additional antigens.

L776 of the tracked version, In terms of preventing acute lung injury and subsequent death due to P. aeruginosa infection, immunization with POmT demonstrated comparable prophylactic levels with PcrV immunization according to all tested parameters, such as specific antibody titers, survival, lung edema, MPO, bacteriology in the lungs, and lung histology.

I thank authors for the explanation about sample size (lines 189-196). However, using different size for mice groups could substantially affect statistics. I recommend authors to consider a priori sample calculation for future studies (see a recent commentary on this in PMID: 33380887).

Response: We thank you very much for your kind suggestion about the consideration for appropriate sample size. We added the following sentence as our explanation why the sample size in groups varies so much in this study.

L303 of the tracked version, It is worth noting that the reasons for the different numbers of animals in each group are as follows. In a single experiment, a maximum of 15 mice were used, including the POmT group as a positive control (effective) group and the alum group as a negative control (ineffective) group, with other groups to be investigated interposed in between to determine the effect. Data were obtained for more than 10 animals/group, with experiments repeated at least three times to confirm reproducibility.

Therefore, in total, the PA103 infection series comprised 42 animals in the POmT group and 28 animals in the alum group, which were both larger than the other groups (10–18 animals).

I think it is confusing for the reader the term “therapeutic” when speaking for vaccines. I consider a therapeutic approach when it is administered to an already infected individual with the objective to treat an infection. In this work all vaccine candidates are injected prophylactically before infection so I strongly recommend to amend the text accordingly.

Response: As the reviewer pointed out, we carefully fixed to use the word “prophylactic”, instead of “therapeutic”.

L37 of the tracked version, an effective therapeutic means… => an effective prophylactic means…

L299 of the tracked version, Therefore, our therapeutic model focuses on the first 4 to 24 h after infection.=> Therefore, our prophylactic model focuses on the first 4 to 24 h after infection.

L672 of the tracked version, Recently, the effectiveness of vaccines (active immunization) with bivalent or trivalent antigens has been investigated [38-41] with the goal of broadening the spectrum of prophylactic efficacy against P. aeruginosa subspecies by targeting multiple virulence factors rather than a single virulence factor.

Round 3

Reviewer 2 Report

The authors have addressed my concerns.